



# Synoptic observations of sediment transport and exchange mechanisms in the turbid Ems estuary: the EDoM campaign

Dirk S. van Maren [1,2,3], Christian Maushake [4], Jan-Willem Mol [5], Daan van Keulen [2,6], Jens Jürges [4], Julia Vroom [2] Henk Schuttelaars[2] Theo Gerkema[7] Kirstin Schulz[8] Thomas H. Badewien[9], Michaela Gerriets[9], Andreas Engels[10], Andreas Wurpts[11] Dennis Oberrecht[11], Andrew J. Manning[2,12,13], Taylor Bailey[14], Lauren Ross[14], Volker Mohrholz[15], Dante M.L. Horemans[16], Marius Becker[17], Dirk Post[10], Charlotte Schmidt [5], Petra J.T. Dankers [18]

[1] State Key Lab of Estuarine and Coastal Research, East China Normal University, Shanghai 200241, China

[2] Delft University of Technology, Delft 2600GA, the Netherlands

[3] Deltares, Marine and Coastal Systems Unit, 2629 HV Delft, the Netherlands

[4] BAW, Federal Waterways Engineering and Research Institute, 22559 Hamburg, Germany

[5] Rijkswaterstaat, 8224 AD Lelystad, the Netherlands

[6] Department of Environmental Sciences, Wageningen University and Research, 6708 PB Wageningen, The Netherlands

[7] Oden Institute for Computational Engineering and Sciences, The University of Texas, Austin, TX 78712-1229, United States

[8] NIOZ Netherlands Institute for Sea Research, Department of Estuarine and Delta Systems, 4400 AC Yerseke, the Netherlands

[9] University of Oldenburg, Institute for Chemistry and Biology of the Marine Environment, Marine Sensor Systems, 26382 Wilhelmshaven, Germany

[10] Niedersächsischer Landesbetrieb für Wasserwirtschaft, Küsten- und Naturschutz, D-26603 Aurich, Germany

[11] NLWKN-Forschungsstelle Küste, Jahnstraße 1, D-26506 Norden, Germany

[12] School of Biological and Marine Sciences, University of Plymouth, Drake Circus, Plymouth, Devon PL4 8AA, United Kingdom

[13] HR Wallingford, Howbery Park, Wallingford OX10 8BA, United Kingdom

[14] Civil & Environmental Engineering, University of Maine, Orono, ME 04469, USA

[15] Leibniz Institute for Baltic Sea Research Warnemünde, D-18119 Rostock, Germany





[16] William & Mary Virginia Institute of Marine Science: Gloucester Point, VA 23062, USA

[17] Kiel University, Otto-Hahn-Platz 1, 24118 Kiel, Germany

[18] Royal Haskoning DHV, 6534 AB Nijmegen, the Netherlands

*Correspondence to*: Dirk S. van Maren (bas.vanmaren@deltares.nl)

**Abstract.**

An extensive field campaign (EDoM) was executed in the Ems estuary, bordering the Netherlands and Germany, aiming at better understanding the mechanisms driving exchange of water and sediments between a relatively exposed outer estuary and
a hyperturbid tidal river. Particularly the reasons for the large up-estuary sediment accumulation rates and the role of the tidal river on the turbidity in the outer estuary were insufficiently understood. The campaign was designed to unravel the hydrodynamic and sedimentary exchange mechanisms, comprising two hydrographic surveys during contrasting environmental conditions using 8 concurrently operating ships and 10 moorings measuring for least one spring-neap tidal cycle. All survey locations were equipped with sensors measuring flow velocity, salinity, and turbidity (and with stationary
ship surveys taking water samples), while some of the survey ships also measured turbulence and sediment settling properties. These observations have provided important new insights into horizontal transport fluxes and density-driven exchange flows, both laterally and longitudinally. An integral analysis of these observations suggest that large-scale residual transport is surprisingly similar during periods of high and low discharge, with higher river discharge resulting in both higher seaward-directed fluxes near the surface and landward-directed fluxes near the bed. Sediment exchange seems to be strongly influenced
by a previously undocumented lateral circulation cell driving residual transport. Vertical density-driven flows in the outer estuary are influenced by variations in river discharge, with a near-bed landward flow being most pronounced in the days following a period with elevated river discharge. The study site is more turbid during winter conditions, when the Estuarine Turbidity Maximum is pushed seaward by river flow, resulting in more pronounced impact of suspended sediments on hydrodynamics. All data collected during the EDoM campaign, but also standard monitoring data (waves, water levels,
discharge, turbidity and salinity) collected by Dutch and German authorities is made publicly available at 4TU Centre for Research Data (https://doi.org/10.4121/c.6056564.v3; van Maren et al., 2022).

# 1 Introduction

The Ems estuary, located on the Dutch-German border, is heavily modified by human activities. Its sediment concentration has increased in the past decades (de Jonge et al., 2014, van Maren et al., 2015a), but the reasons for this increase are still
under debate. The outer Ems estuary is connected to the lower Ems River (see Figure 1) which has a fairly low discharge but





does not, or only very limitedly, supply sediments. A tributary system (Leda-Jümme-basin) that accounts for approximately 1/3 of the tidal volume of the lower Ems River drains a peat bog, thereby providing a considerable amount of humic acids and other organic material. The present-day lower Ems River is characterized by thick and mobile fluid mud with concentrations up to 200 kg/m³ (Papenmeier et al., 2013) which migrates up- and down-estuary with the tide over a distance of about 10 km.

During low river discharge conditions high sediment concentrations are measured up to the tidal limit, the weir at Herbrum (Talke et al., 2009). In order to keep the lower Ems River navigable, 1 to 1.5 million ton is annually extracted from the lower Ems River by dredging (Vroom et al., 2022). The fluvial Ems River does not carry a substantial sediment load. Most likely this sediment is of marine origin transported up-estuary by the tides (Chernetsky et al., 2009; van Maren et al. 2015b; Dijkstra et al., 2019a) although currently the contribution of particulate organic matter (rather than inorganic matter) released by the

Leda-Jümme-basin is also being investigated.

The suspended sediment concentration (SSC) in the lower Ems River has increased much more than in the outer estuary (de Jonge et al., 2014). The river became hyper-turbid sometime during the 1990's, most likely between 1989 and 1995 (Dijkstra et al., 2019b). This transition towards hyperturbidity is probably related to channel deepening influencing the tidal dynamics and the sediment concentration through a positive feedback mechanism introduced by Winterwerp et al. (2013). Initial

deepening led to more tide-induced sediment import, which resulted in a turbulence damping and thereby a lower apparent hydraulic roughness, amplifying the tides and further strengthening sediment import (Winterwerp and Wang, 2013; Winterwerp et al., 2013; van Maren et al., 2015b, Dijkstra et al., 2019a,b). Tidal amplification has been additionally influenced by the upstream weir at Herbrum (Schuttelaars et al., 2013) whereas sediment supply may have been influenced by changing dredging activities in the beginning of the 1990's (van Maren et al., 2015a). The tidal up-estuary transport mechanisms appear

to be a combination of the spatial settling lag (Chernetsky et al., 2010) and mixing asymmetry (Winterwerp, 2011).

Our understanding of the sediment dynamics in the outer estuary and lower Ems River is primarily based on mathematical modelling and only limitedly based on in situ observations. Partly resulting from the absence of data, a number of key questions remain fuelled by lack of data or disagreement between earlier work. The first is that while transport mechanisms in the lower Ems River has been studied in great detail (see references above) less is known about the transport mechanisms in the channel

connecting the lower Ems River and outer Ems estuary (the Emden Navigation Channel or ENC). The tides in the ENC are asymmetric with higher ebb flow velocities than flood flow velocities (Pein et al., 2014), and even though models result that spatial asymmetries are more important for residual transport than temporal asymmetries (Chernetsky et al., 2010) these tidal dynamics need to be explored in more detail. Also, salinity-driven flows appear insufficiently strong to import large quantities of suspended sediments landward (van Maren et al., 2015a). Secondly, some of the sediment in the lower Ems river is flushed

seaward during periods of elevated discharge. The accumulation of sediment ion the lower Ems river may therefore lead to an increase in turbidity in the outer estuary. However, to what extent the high turbidity in the lower Ems river influences the turbidity in the outer Ems estuary (for instance during flushing) remains poorly known. And thirdly, the ENC requires large amounts of maintenance dredging, whereas from a hydrodynamic point of view it is one of the most energetic sections of the



estuary. This points to the presence of convergence of suspended sediment transport, which in turn conflicts with the large

dredging volumes in the lower Ems River suggesting up-estuary transport.

Answering these three questions requires a better understanding of the exchange mechanisms between the estuary and the lower Ems River, especially within the ENC. For this purpose, a large-scale field observation campaign involving eight research vessels was carried out (the Ems-Dollard Measurements or EDoM campaign). The aim of this paper is to document the data collection during this campaign and draw major conclusions based on an initial analysis of the data. Section 2 details

the design phase of the campaign, based on a review of exchange mechanisms between the outer Ems estuary and the lower Ems River. This review translates into a detailed methodology described in section 3. The actual deployment conditions and some key observations are described in section 4. These findings are discussed and used to address the three research questions formulated above.

## 2 Design of the experiment: exchange processes between the outer Ems estuary and lower Ems River

### 2.1 Up-estuary transport mechanisms and the ETM

Regions of elevated sediment concentration within an estuary are referred to as Estuarine Turbidity Maxima (ETM's), and result from converging sediment transport mechanisms (see e.g. Burchard et al., 2017). These converging pathways are driven by river flow, estuarine circulation and lag effects (Dyer, 1994). Estuarine circulation is the combined effect of gravitational circulation (Postma, 1967), internal tidal asymmetry (Jay and Musiak, 1994) due to tidal straining (Simpson et al., 1990),

lateral tidal residual flows (Lerczak and Geyer, 2004) and river flow; the relative importance of each component is strongly site-specific and often variable in time. Residual transport by time lag effects (such as settling lag and scour lag) are the result of finite values of the sediment properties (settling velocity, critical shear stress for erosion) in combination with asymmetries in the hydrodynamics (time or spatial asymmetries). Sediment properties vary throughout the tidal cycle, and – especially at high sediment concentrations – influence the hydrodynamics (diffusivity, viscosity).

Most sediment in the lower Ems River is of marine origin, as the Ems River itself does not transport substantial amounts of sediments. The pronounced ETM in the lower Ems River is therefore primarily transported up-estuary by marine processes. Approximately 1 to 1.5 million ton is annually extracted from the lower Ems River by dredging (Vroom et al., 2022) and assuming that the sediment mass in the Ems River (in suspension and in the bed) does not decrease, the up-estuary residual flux is at least 1 to 1.5 million ton/year. The ETM used to be located near the tip of the salt wedge (de Jonge et al., 2014) but

fluid mud is presently observed many kilometres landward of the salt intrusion limit (Talke et al., 2009). This transport of sediment up-estuary of the salt limit may be the result of sediment-induced density currents (Talke et al., 2009), tidal asymmetry (Chernetsky et al., 2010; van Maren, 2015b), or lag effects (Chernetsky et al., 2010), although a combination of these processes seems most likely (Dijkstra et al., 2019b). Down-estuary of the turbidity maximum (in the outer Ems estuary) the tides are more symmetrical (although still flood-dominant; Pein et al., 2014), and salinity-driven gravitational circulation

(van Maren et al., 2015b) and tide-induced residual flows (van de Kreeke and Robazewska, 1993) also contribute to residual



sediment transport (van Maren et al., 2015b). Processes which may additionally influence residual sediment transport are flocculation and/or sediment-fluid interactions. It was hypothesized by Winterwerp (2011) that tidal asymmetries in flocculation lead to a pronounced up-estuary sediment transport. Sediment-fluid interactions are known to influence sediment transport within the lower Ems River (Talke et al., 2009; Winterwerp, 2011; van Maren et al., 2015b; Becker et al., 2018), but

it is not known to what extent these density-induced effects also influence turbulent mixing, salinity stratification and sediment dynamics in the ENC.

### 2.2 Impact on the outer Ems estuary

Although the residual sediment transport is directed from the outer Ems estuary to the lower Ems River (resulting in regular dredging of the lower Ems River), sediment may also be transported from the lower Ems River to the outer estuary. There are

indications that such seaward transport takes place during high discharge events (Spingat and Oumeraci, 2000; van Maren et al., 2015b). However, it may also be that the tides are so flood-dominant that the reduction in flood flow velocities during high discharge events leads to a reduction in the maximal bed shear stress, leading to consolidation and hence sediment trapping in the upper reaches of the lower Ems River (Winterwerp et al., 2017). Understanding the effect of river discharge on sediment dynamics requires detailed observations of sediment transport parameters during or shortly after (which is logistically more

feasible) high and low river discharge.

A second mechanism through which the high sediment concentration in the lower Ems River influences concentrations in the outer Ems estuary is shear dispersion. Mixing of a lateral concentration gradient by tidal currents generates a net transport flux that is proportional to the concentration gradient (and directed to the area with the lowest sediment concentration, i.e. the outer Ems estuary). Quantifying the flux by shear dispersion requires knowledge of the horizontal concentration gradient.

**2.3 Storage in the ENC**

The transition between the lower Ems River and the outer Ems estuary is sheltered by the Geise Dam. The larger part of this ~12 km long section is also the approach channel to the port of Emden (the ENC), which is dredged to -10.5 meter below mean sea level to provide access to the port. This length is close to the tidal excursion, and therefore the water-bed interaction in this area is important for exchange processes between the lower Ems River and the Ems estuary. In terms of flow velocities, this

region is one of the most dynamic areas of the total Ems estuary (Pein et al., 2014). But, despite these high flow velocities, approximately 1.6 million tons (3.2 million $m^3$) of fine-grained sediments are annually dredged from the ENC. This suggests that the ENC is a zone where sediment transport pathways converge (e.g. seaward flushing from the Ems River and up-estuary transport by tidal pumping or estuarine circulation).

With a large amount of mud that is regularly resuspended, it is likely that mobile, highly concentrated near-bed suspensions exist. Sediment particles in such suspensions settle slowly because of hindered settling effects. Since the material is regularly

vertically mixed, there is insufficient time to develop into a fluid mud or solid bed. Such suspensions have a density in-between that of a fluid mud (several tens to 100's $kg/m^3$) and a suspension (several 0.1 to 1 $kg/m^3$). Such high-density layers influence



the turbulence structure of the water column, generating stratification (and thereby influencing sediment transport mechanisms), but have importance also to sediment-induced density currents. Such high-concentration suspensions are common in the Ems River (Talke et al, 2009), but to what extent they also exist in the ENC is unknown.

## 2.4 Design of the campaign

It is likely that salinity-driven residual flows and mixing/stratification influence sediment exchange. The measurements should therefore cover a period with high river discharge and a period with a low river discharge. Observations should include the vertical structure of the water column (salinity, SSC, velocity) covering a wide spatial scale (the lower Ems River, the outer Ems estuary, and positions in-between) and temporal scale (to account for subtidal variations in water level and river discharge). The vertical structure of the water column requires boat surveys (equipped with ADCP and CTD) whereas the large timescales require frame observations (which do not cover the vertical structure of the water column). As an alternative, a series of frames are deployed to measure for a period of at least one spring-neap cycle) while 13-hrs boat surveys (profiling the water column) are executed during the period of frame measurements. All stations include observations of flow velocity, salinity and SSC. At some stations, these observations are supplemented with observations of turbulence, settling velocity, or particle size.

## 3 Deployment

In order to capture the contrasting conditions resulting from the river discharge, two measurement campaigns were defined: one in August 2018 (local summer with relatively low river discharge) and one in January 2019 (beginning of the wet winter conditions). From a physical point of view a later winter deployment was preferred (longer duration of larger freshwater flow) but intense maintenance dredging and operations of a storm surge barrier planned in February – March imposed the winter campaign to be executed in January. Acquiring a synoptic pattern of flow and transport patterns required the deployment of a large number of observation stations, which motivated the collaboration of many governmental and scientific institutes and universities, each deploying their own equipment and/or research vessel. Long timeseries (of at least one spring-neap cycle) were collected using moorings to cover the temporal variation in transport processes while simultaneous short-term (13-hours) deployments were executed to investigate detailed processes in the vertical (through profiling of salinity, temperature, turbidity and for some stations turbulence and floc properties) or over the cross-section (using ADCPs). The collected data set was complemented with permanent observations already executed as part of existing monitoring frameworks. The permanent observations, spring-neap observations, and 13-hrs observations will be explained in more detail hereafter.



### 3.1 Instrumentation

A large number of permanent observation stations are available in the Ems estuary, measuring water levels, SSC/ salinity/oxygen/temperature or wave height. In order to relate conditions during the EDoM campaign to long-term environmental conditions, a selection of observations collected at the permanent monitoring stations is added to the EDoM dataset over the period July 2017 – June 2019. The top panel in Figure 1 provides an overview of the locations where long-term monitoring data is available. The offshore station Randzelgat Noord measures wave data, while stations Knock, Emden, Pogum, Gandersum and Terborg measure turbidity, salinity, oxygen content and temperature. Turbidity is converted to SSC (with values up to several 10's of kg/m$^3$) through calibration curves. For stations Knock, Emden and Terborg water levels are additionally provided. The river discharge is measured at Versen, 40 kilometer upstream of the weir at Herbrum (and ~100 kilometer upstream of Emden).

Observations covering at least one common spring-neap tidal cycle were executed at so-called bottom-mounts (BM), mooring chains (MC) and larger bottom frames (RS) – see Table 1. In total 5 bottom mounts, 3 mooring chains, and 2 larger frames were deployed throughout the study area (Figure 1). The bottom mounts were equipped with an upward-looking 600 mHz TRDI WH-S ADCP and a RBR concerto CTD with Seapoint STM turbiditymeter sensor at 0.5 m.a.b. to measure salinity, turbidity and temperature (Table 2). The mooring chains measured salinity, turbidity and flow velocity at a height of 1.5, 3.5, and 7.7-7.9 m.a.b. (depending on location, see Table 2). The large frames measured flow velocity throughout the water column by combining an upward-looking 600 mHz TRDI WH-S ADCP and a downward-looking high resolution Nortek Aquadopp HR velocity profiler. Salinity, temperature, and turbidity was measured using OBS 3A's deployed 0.2, 0.3, 0.5, and 0.8 m.a.b..

During each campaign 8 ships measured simultaneously in three different modes. Four ships were deployed in stationary mode, with SB_KNO, SB_EFW and SB_EMD remaining anchored throughout the tidal period while SB_POG floated with the currents. All surveys started half an hour before local low water and ended half an hour after local high water (Figure 2). Slack tide is close to high and low water in the Ems Estuary, and therefore the observations covered the period from low water to low water, but also from low water slack to low water slack.

All anchored ships measured the flow velocity with a downward-looking ADCP and temperature and salinity with a Conductivity-Temperature-Depth (CTD) profiler equipped with an Optical Backscatter Sensor (OBS). Water samples were taken at least once every hour at three water depths (near-surface, near-bed and halfway the water column except for SB_POG which measured near bed and near surface only because of shallow waters) and analysed in the laboratory for suspended sediment mass. Water samples are, in general, important for conversion of turbidity (measured by the OBS) to SSC. However, with the high concentrations in the Ems estuary acoustic and optical instruments become progressively less reliable, making



accurate water sampling an important source of data. In order to minimise the potential impact of methodological errors all water samples were therefore analysed in the same laboratory.

Additional instrumentation was deployed onboard the stationary boats at Emden (SB_EMD) and Knock (SB_KNO) to measure turbulence and sediment settling properties. At SB_EMD, hourly water samples were taken with a Niskin bottle close to the bed and close to the water surface. A subsample taken with a pipette is inserted into a still and clear water settling column operated onboard, in which the water-sediment mixture settles from suspension. This settling is monitored with a high-

resolution video camera. Postprocessing of the camera data reveals the size, shape, and settling velocity of all particles registered with the camera. This provides a population of settling speeds and floc sizes, which can be averaged into a sample-averaged value (see e.g. Manning and Dyer, 2002). At SB_KNO, the settling properties were measured in August 2018 with a LISST200x in profiling mode using a high turbidity module. Turbulence was measured at the location of SB_EMD using a Rockland Scientific MicroCTD which was profiling the water column from a free-floating small vessel close to SB_EMD.

The MicroCTD was used to collect measurements of turbulent kinetic energy (TKE) dissipation, as well as salinity, temperature, and turbidity data over 13h transects during the summer (August 28, 2018) and winter (January 24, 2019) field campaigns. Data were collected in a series of vertical casts, with approximately 5 casts performed every 15 minutes. The 5 casts were averaged and bootstrapped 6,000 times to provide a statistically significant measurement of TKE with depth (Efron & Gong, 1983; Huguenard et al., 2019; Ross et al. 2019 and references therein). Turbulence and flocculation properties were

also measured at SB_KNO in August 2018 using an MSS-90-S Microstructure Profiler sampling at 1024 Hz. The profiler was used in free fall mode with a downward velocity of approximately 0.6 m/s. The TKE dissipation rate was calculated by fitting the observed shear spectrum to the theoretical Nasmyth spectrum in a wave number range from 2 to maximum 30 cycles per meter.

The cross-sectional profiles were only deploying ADCPs except for CS_POG which towed a FerryBox (2018) or CTD (2019). Transects were continuously sailing back and forth over a GPS-steered track to cover the mouth of the Dollard (CS_DOL), the Emden fairway (CS_EFW) and the lower Ems River (CS_POG). The backscatter of the ADCP was calibrated to SSC using water samples collected at nearby stationary boats. A longitudinal survey was carried out to measure near-surface salinity and SSC, sailing landward during the flood period (from Borkum on the island of Norderney to Papenburg close to the landward

limit of the lower Ems River) and back during the following ebb period.

**3.2 Data processing**

All data was carefully examined for outliers and spikes and, for the stationary surveys the OBS's were calibrated to physical units (SSC). The data uniformly collected over all stations (velocity, salinity, SSC) was subsequently averaged to a 10-minute time interval (and stored on a data repository). Additional datasets (concentrations and settling velocity from water samples,

LISST data, turbulence data) is not averaged or averaged over a different time period or over a number of samples.



Additional data processing depends on the purpose of the data analyses. In this paper we provide a synoptic view of residual flow and sediment transport, involving the conversion of point observation data into fluxes. Residual fluxes can be computed from

- 13-hrs stationary ship observations (resolving the vertical variation in the flow velocity and SSC (based on OBS profiling), but lacking cross-sectional variation and resolving only a short period of time),
- 13-hrs transect observations (resolving the cross-sectional variation but using the ADCP's echo-intensity for SSC and resolving only a short period of time)
- Spring-neap observations using moored instruments (providing a much longer averaging period, but using near-bed OBS observations for SSC and lacking cross-sectional variation in the flow and SSC)

For reasons elaborated in section 4, we will use the spring-neap observations to compute fluxes. For the RS and BM frames the time-varying point fluxes $F_p$ are defined as in Eq. 1

$$F_p = \bar{u} c_b h \qquad (1)$$

where $\bar{u}$ is the depth-averaged ADCP velocity profile, $c_b$ is the concentration measured at 0.8 m (RS frames) or 0.5 m (BM frames), $h$ is the water depth. For the MC stations (measuring at three positions in the vertical) the average of the product of the $u$ and $c$ (measured at height $i$) is multiplied with the water depth as in $F_p = \frac{h}{3}\sum_{i=1}^{3} u_i c_i$. For all observation locations, $F_p$ is converted into a tide-averaged flux by integrating over a spring-neap tidal cycle ($T = 14.77$ days), dividing by the number of $M_2$ tidal cycles $N$ (with $N = 28.54$), and multiplying with the channel width $W$:

$$F = \frac{W}{N} \int_0^T F_p(t) dt. \qquad (2)$$

We realise that this method has several shortcomings. Multiplying the depth-averaged velocity profile with a concentration measured at $0.6 - 0.8$ meter above the bed leads to an overestimation of the total flux. Additionally, multiplying a point measurement with the channel width ignores cross-channel variabilities in residual flow and SSC. We will revisit these shortcomings later in this manuscript.

### 3.3 Environmental conditions

The 2018 surveys were characterised by low wind speed, wave height and river discharge, i.e. representing tide-only conditions (Figure 3) – see also Schulz et al., 2020. The river discharge during the 2019 surveys was higher than in 2018 although rather low for winter conditions. The first high river discharge peaks occurred relatively late in the season, with the 13-hour measurements exactly in-between two high discharge events. Offshore wave conditions were slightly higher in winter than in summer, without prominent storm conditions.



The difference in discharge conditions resulted in a markedly different salt intrusion and location of the Estuarine Turbidity
Maximum (ETM) (Figure 4). In summer, the salinity in the fairway to Emden (grey shaded in Figure 4a) is between 20 and 27
ppt, and the ETM is completely located in the lower Ems River (Figure 4b). During winter conditions, the salinity in the
fairway is between 5 and 20 ppt (Figure 4c). The ETM has shifted 25 km in the seaward direction and is partly located in the
fairway to Emden (Figure 4d).

## 4 Transport mechanisms

### 4.1 Residual flow

Horizontal residual flows can be computed from the frames, stationary boats, and from boats sailing in transects. Figure 5
displays the tide-averaged residual flow from the 13-hrs stations as well as the longer moored instruments near the surface and
near the bed, on 28 August 2018 and 24 January 2019. The combined point and transect observations reveal a consistent pattern
of residual flows. In the mouth of the Dollard, the transect data reveal cross-sectionally varying residual flows, especially near
the surface, but averaged over the cross-section there is no preferential inflow or outflow. Hence the moored observation
stations in the Dollard (BM_DOL and MC_DOL) represent the flows through the mouth of the Dollard. Station RS_DOL
reveals net outflow (both near the surface and near the bed), probably resulting from local bathymetric constraints.
In the mouth of the fairway to Emden a pronounced cross-channel and vertical stratification pattern exists during both low
discharge (Figure 5a, c) and high discharge conditions (Figure 5b, d). The surface flow velocities are directed seawards (Figure
5a, b) whereas the near-bed currents are directed landwards (Figure 5b, d) . On top of this, a pronounced south to north gradient
exists, with prevailing landward residual flow in the South and seaward flow in the North. Apparently, the velocities in the
moored stations in the North are slightly seaward-directed while those in the south are more landward-directed. This pattern
will be elaborated in more detail hereafter.

During the ebb, the along-channel flow velocities are much larger along the northern outer bend (cross-section CS_EFW in
Figure 6d); we attribute this to bathymetric constraints imposed by the curved channel. During the flood, however, the flow is
much more cross-sectionally uniform. Averaged over the tidal cycle, this leads to outflow along the northern bend and inflow
along the southern bend (see cross-section CS_EFW and station RS_EFW in Figure 5). This cross-sectional variation of the
along-channel flow velocity also gives rise to a transverse flow pattern with a northward directed bottom current during both
the ebb and the flood (Figure 6b, e), compensated by a southward flow near-surface. A curvature-induced secondary flow
would lead to a near-bed flow from the outer bend to the inner bend, i.e. towards the south during both the ebb and flood. The
northward near-bed flow can be explained, however, by the lateral advection of the salinity gradient. During the ebb, the cross-
sectional variation of the flow leads to a lower salinity along the northern bend compared to the southern bed. This positive
salinity gradient from North to South drives a classic gravitation flow with northward flow near the bed and southward flow
near the surface. During the following flood phase, this cross-sectional salinity gradient is largely maintained because the flood



currents are cross-sectionally uniform. Therefore, the near bed transverse currents remain directed towards the north (and towards the south near the surface).

Stratified flows do not only exist perpendicular to the main flow direction (as shown in Figure 6) but also in the along-channel direction. These along-channel residual flows are revealed by low-pass filtering the along-channel flow (Figure 7). All stations reveal a seaward-directed residual surface current during periods of high river discharge, and a landward-directed near-bed current during the waning stage of the river discharge peak. This discharge dependency appears to be weaker for stations within the ENC (GEI and EFW) than for those in the Dollard and outer Ems estuary (DOL and KNO). Such near-bed flows are important for residual sediment transport (as typically most sediment transport takes place close to the bed) but unfortunately the lowest 2 meters of the water column are not measured with the upward-looking ADCPs. It is therefore likely that landward flows are more strongly developed than suggested by Figure 7.

## 4.2 Residual sediment transport

The residual sediment transport can be computed with the transect observations, the moored 13-hrs stations, and the moored instruments (see section 3.4). Figure 7 reveals that the subtidal flows are substantially varying, especially during the January surveys, in response to river discharge fluctuations. This discharge variability not only influences the hydrodynamics but also the supply of sediments from the lower Ems River (with a high river discharge flushing sediments seawards). Because of this variability the 13-hrs surveys may not represent typical conditions (especially during the winter observations). We therefore present residual fluxes using the moored instruments (Figure 8) measuring for a spring-neap tidal cycle.

The computed fluxes using all moored instruments provide a spatially, temporally, and vertically consistent picture. The three Dollard moorings yield fluxes in the same direction but even of comparable magnitude within each campaign and between campaigns. These observations strongly suggest the Dollard is importing sediments. These residual fluxes are not the result of residual flows (Figure 5) but of an asymmetric availability of sediments (Figure 9). The SSC is higher during the flood than during the ebb, probably at least partly consisting of sediment that was discharged into the outer Ems Estuary from the lower Ems River during the previous ebb phase.

The fluxes in the ENC are directed seaward, during both low and high discharge conditions, and on both the north bank (BM_GEI, RS_EFW) and south bank (BM_EFW and MC_EFW). The latter is important given the cross-sectional variability in longitudinal flows (Figure 6): apparently the residual transport is directed seaward despite periods with landward directed residual flows. The seasonal coherence in residual fluxes is surprising, especially given the salinity-driven landward currents in response to periods with higher river discharge revealed in Figure 7. The higher river discharge in January does lead to a larger difference between near-bed and near-surface transport: in general the (seaward-directed) surface sediment fluxes are larger in January than in August while the near-bed fluxes are weaker (or even landward directed, as for RS_EFW).





The consistency of the computed sediment fluxes is further supported by an evaluation of the gross and net sediment fluxes
(Figure 10). Systematic differences between ebb and flood fluxes may easily arise from bathymetric constraints (the
measurement location is in a flood or ebb-dominant location) or measurement shortcoming (a slightly different sediment type
during ebb or flood leads to differences in ebb and flood concentrations due to the dependence of sensors to the sediment grain
size). A net sediment flux of e.g. 1% may then reverse direction with a 2% error in the gross fluxes. However, in the Dollard,
the residual flux is approximately 20% of the gross flux. Such a large net flux (relative to the gross flux) makes it relatively
insensitive to measurement shortcomings. In the ENC the ratio of net to gross fluxes is more variable (in-between several %
at RS_EFW and 40% at BM_EFW) but overall, typically more than 10%. This suggests that also the fluxes in the fairway to
Emden are fairly accurate.

Station BM_KNO reveals a pronounced landward residual transport in August but was unfortunately malfunctioning in
January (Figure 8). This residual sediment transport is in agreement with large up-estuary transport illustrated with the dredging
volumes in the ENC and lower Ems River as well as the hyper-turbid condition in the lower Ems River. This landward transport
results from a phase difference between maximal flow velocity and maximal sediment concentration, illustrated in Figure 11.
The first half of the flood is characterized by a faster rise of water levels compared to the second half, whereas the falling stage
is much more uniform – the duration of rising and falling water levels is the same (Figure 11a). As a result, (1) the flood flow
velocities are maximal at the beginning of flood, whereas ebb flow velocities are maximal later in the ebb and (2) the flow
velocity peaks are slightly higher than the ebb velocity peaks (Figure 11b). Although depicted here for station BM_KNO, this
asymmetry in velocity peak phasing is observed throughout the various observation stations, although in some stations ebb
flow velocities are higher. An asymmetry with a different duration of the period of high water slack compared to low water
slack is known as slack duration asymmetry (Friedrichs, 2011). For a longer duration of high water slack (as at station
BM_KNO) is typical for high water duration asymmetry, with a water level phase difference $\theta_\zeta$ between the M2 and M4 tidal
constituents $\theta_\zeta = 2\phi_{\xi_{M2}} - \phi_{\zeta_{M4}}$ between 90 and 270 degrees (and maximal at 180º). This is supported by long term water
level observations collected at Pogum, of which tidal analysis reveals a value for $\theta_\zeta$ very close to 180º (van Maren et al., 2015).
The observed tidal asymmetry in SSC (Figure 11c) is primarily reflecting advection of sediment flowing out of the ENC at the
end of ebb (18:00 – 20:00) which is after flow reversal transported SSC back into the ENC (20:00 – 21:00 and 8:00 – 10:00).
However, during the flood a second SSC peak exists (from 8:30 to 9:30) superimposed on the advection peak and
corresponding to the flow velocity peaks. The cumulative sediment fluxes (Figure 11d) show that this phase (where a period
of high SSC coincides with a period of high flow velocity) has a major influence on the residual sediment transport. Apparently,
the early peak in the flood flow velocity (resulting from high water duration asymmetry) is important for the residual transport
of sediment. The observed residual transport in Figure 11 is also representative for a longer period of time: the computed
cumulative flux of $5 \cdot 10^3$ kg/m² over the tidal cycle corresponds to 0.11 kg/m²/s which is very close to the average flux computed
over a spring-neap tidal cycle (Figure 8). Spatially, however, the direction of residual fluxes is more variable. Transport is ebb-





dominant in station SB_KNO (see Schulz et al., 2020), located 1 km SE of BM_KNO. This difference may reflect lateral variability in the plume flowing out of the lower Ems River, directly crossing location SB_KNO during the ebb but then deflecting northward to travel past location BM_KNO during the flood.


The observations in the mouth of the Dollard show a remarkable similarity with those collected at Knock (Figure 12). The flood flow velocities peak at the beginning of flood which, combined with the high SSC during this period, results in a large landward directed sediment flux. It seems likely, however, that the large SSC peak measured at the beginning of flood is especially large in the mouth of the Dollard because turbid water that was discharged from the ENC during the previous ebb

flows into the Dollard with the incoming flood currents. Despite the large landward sediment flux (10 $10^3$ ton / tide corresponding to 7 million ton/year), bathymetric data records reveal there is no net accumulation of sediment in the Dollard. Apparently, sediment must also flow out of the Dollard – this will be discussed in section 5.

High water duration asymmetry provides a mechanism transporting sediment into the outer estuary (and partly into the

Dollard), but this asymmetry is insufficient to explain the large sediment flux towards the lower Ems River. The strongest evidence for mechanisms driving transport from the ENC to the lower Ems River is provided by water samples collected at the beginning of the ENC (SB_EFW), halfway into the ENC (SB_EMD) and within the lower Ems River (SB_POG). Water samples provide an important methodology to measure the sediment concentrations in the ENC because of the high SSC (up to 35 kg/m$^3$, which is beyond the detection limit of many optic and acoustic instruments), and because of technical difficulties

with the POG observations. Simply comparing the SSC concentration of these three stations throughout the 13-hr tidal cycle sampled in January 2019 reveals a progressive increase in near-bed SSC during the flood from <1 kg/m$^3$ (EFW) to ~15 kg/m$^3$ (EMD) to ~30 kg/m$^3$ (POG). This spatial trajectory is within the tidal excursion (<15 km) and therefore the most likely explanation of this landward increase in SSC is sediment resuspension from the bed. The origin of this sediment will be discussed in more detail in the following section. The concentration is much lower during the following ebb (~10 kg/m$^3$)

suggesting the sediments transported landward during the flood have deposited in the lower Ems River. Interestingly, during this period of apparent strong sediment import the salinity profiles were reversed (with higher salinity values near the bed than near the surface), suggesting partly decoupled flow dynamics in the highly concentrated layers near the bed from the water masses higher in the water column (as has been described for the fluid mud reach in the lower Ems river by Becker et al., 2018).

**4.3 Mixing and flocculation**

Turbulence data was collected at SB_KNO (August 2018 only) and at SB_EMD (both campaigns). The SB_KNO turbulence data provide insight in mixing and stratification processes in response to lateral and longitudinal flows (Schulz et al., 2020) whereas the SB_EMD data reveal how sediment-induced stratification processes may promote ebb-dominant sediment transport (Bailey et al., in prep.). The floc size measurements reveal a large variability in settling velocity within a tidal cycle



(with higher settling velocities during the flood than during the ebb, but also a large seasonal variability: the settling velocity was higher during the August observations than during the January observations.

## 5 Main findings and future work

The three research questions that motivated this study where related to the mechanisms for upstream sediment transport, the
impact of high SSC in the lower Ems River on the outer estuary, and the high maintenance dredging rates in the ENC. This EDoM campaign has provided important new insights into the mechanisms regulating exchange of water and sediment between the lower Ems River and its outer estuary. At the same time these new insights have also raised new questions that need to be addressed, partly through more detailed analyses of the collected data. We will first address the main questions motivating the measurement campaign, followed by potential follow-up studies using the collected data.

**5.1 Sediment dynamics.**

### 5.1.1 Landward sediment transport mechanisms

One of the motivations for the EDoM campaign was to understand the mechanisms leading to landward transport in the ENC, because known mechanisms (tidal asymmetry, salinity-induced estuarine circulation) appeared to be too weak to explain the large landward transport. Surprisingly, the field observations suggest that the ENC is exporting whereas the Dollard is
importing. It is therefore hypothesized that a large-scale horizontal circulation exists where sediment flows into the Dollard and flows back into the ENC during either fair-weather conditions or during storm conditions. A substantial residual flow over the Geise dam from the Dollard to the ENC has indeed been observed (Jensen et al., 2002). However, subsequent observations carried out to further validate this (not reported here) have not yet confirmed this residual transport due to higher SSC in the ENC than in the Dollard. The role of the Dollard, and the closure of the mass balance in the Ems Estuary (Figure 8) is therefore
still not completely resolved.

Based on single station observations, the main mechanism responsible for the sediment transport towards the ENC and the Dollard appears to be slack water duration asymmetry. Further landward spatial asymmetries become progressively more important. Transport is directed towards the Dollard because of the high SSC at the beginning of the flood, which can be traced back to outflow of turbid water from the ENC but also explained by (high water) slack water duration asymmetry. However,
the concentration peak at the beginning of the flood (leading to overall flood-dominant transport during calm conditions) was already observed in 1996 (Dyer et al., 2000) when the lower Ems River was not as turbid as it is nowadays. This suggests that transport into the Dollard is primarily driven by slack high water slack duration asymmetry, strengthened by outflow of turbid water from the ENC.

Net transport from the ENC into the lower Ems River is not driven by an asymmetry in the flow, but by sediment availability.
The large role of sediment availability is demonstrated by the steep increase of the landward sediment flux throughout the



ENC (Figure 13). This large sediment availability may reflect sediment transport from the outer Ems estuary via the Dollard, over the Geise dam into the ENC (as discussed above). Alternatively, the large up-river sediment flux may be the result of large sediment deposits during a high discharge event that occurred several days before the 13-hrs observations (Figure 7).

**5.1.2 Impact of high SSC in the lower Ems River on the outer Ems estuary**

Prior to the measurement campaign, sediment was believed to be transported up-estuary (through the ENC to the lower Ems River) during low discharge conditions, being flushed out again during periods of higher river discharge (Spingat and Oumeraci, 2000; van Maren et al., 2015b). The outer Ems estuary may therefore be more strongly impacted during high discharge conditions. In addition to the river discharge, the high sediment concentrations in the lower Ems River permanently increase the sediment dynamics in the downstream outer estuary by shear dispersion.

The collected dataset suggests a number of mechanisms exist that reduce the effect of the lower Ems River on the outer Ems estuary. First, high discharge events are immediately followed by a phase of intensified gravitational circulation, with a landward directed current transporting sediment that settled from suspension after the period of higher discharge back towards the ENC. Secondly, the asymmetry of tidal currents (with a high flow velocity at the beginning of flood) suggest that turbid water flowing out of the ENC is effectively transported landward. And thirdly, the sediment-rich water flowing out of the ENC
is diverted back into the Dollard, from which the sediment is hypothesized to flow back into the ENC over the Geise dam. On the other hand, two mechanisms have been identified which could raise SSC in the outer Ems estuary due to the high turbidity in the lower Ems River. First, the sediment transport in the ENC is directed seaward, providing a permanent conduit for sediment to be transported from the lower Ems River into the outer Ems estuary. Secondly, part of the turbid water flowing out of the ENC directly flows into the Dollard after reversal of the tides. This would suggest a steady increase in SSC in the
Dollard in response to the increasing SSC in the lower Ems River, which is in line with observations (van Maren et al., 2015a). Determining which of the mechanisms above is stronger, and hence to what extent flushing of the lower Ems River influences the turbidity in the outer Ems estuary, requires additional modelling. The EDoM dataset provides new insights on processes to resolve in such models as well as observations to calibrate the models with.

**5.1.3 Large maintenance dredging rates**

The computed sediment fluxes suggest convergence of sediment transport at the mouth of the ENC, which is exactly the location where most maintenance dredging is taking place. Prior to the measurement campaign the reasons for this location where poorly understood, as the ENC was considered to be transporting sediment from the outer Ems estuary towards the lower Ems River. However, the seaward sediment transport suggested by the moored observations provide a good explanation for the location and the magnitude of the maintenance dredging rates. Based on a preliminary analysis of the collected data,
the convergence of sediment transport appears to be the result of a balance between (1) high water slack asymmetry (driving a landward transport in the estuary) and (2) the high concentration gradient and a seaward-directed residual flow in the ENC (driving the seaward sediment flux).



## 5.2 Future work

The EDoM measurement campaign provided important new insights into sediment dynamics, but also exposed gaps in
knowledge that may be filled in by additional analyses of the collected data. We therefore additionally provide directions for
future research using the collected dataset.

- The collected synoptic dataset has substantially increased our understanding of exchange mechanisms between the
outer Ems estuary and the lower Ems River. However, the relative importance of these mechanisms on exchange,
including their seasonal variability, still remains to be quantified in more detail. A way forward here is to
decompose the sediment fluxes into tidal pumping, advection, and estuarine circulation terms (following e.g. Dyer,
1988).
- The computed sediment fluxes provide a consistent picture of the residual transport, but their accuracy could be
improved because of assumption made to vertically and cross-sectionally extrapolate the data. The cross-sectional
variation of the flow (as measured at the transects) and the vertical distribution of SSC (measured with the
stationary boat surveys) provide means to better extrapolate the fluxes over depth and the channel cross-section.
- The measurements in the Dollard and ENC suggest that a horizontal residual transport cell (with sediment transport
towards the Dollard and via the Geise dam into the ENC) exists, but this circulation has not yet been supported by
an equivalent transport magnitude over the dam based on field data. It is recommended to further investigate this
circulation pattern through a combination of modelling work, collection of new data, and/or re-analysis of the
sediment fluxes (as described above).
- Slack water duration asymmetry appears to be an important mechanism transporting sediment landward in the outer
Ems estuary. Its importance for residual landward transport should be investigated in more detail through
systematic tidal analyses of water levels and flow velocity, and the intra-tidal relation between currents and SSC.
- A first analysis of the flocculation data (not shown here) has indicated that the intra-tidal and seasonal variability in
the settling velocity is large and may contribute to sediment deposition in the ENC, and therefore the seasonal
variation in maintenance dredging volumes.
- The sediment concentration gradients in the ENC were sufficiently large to influence turbulent mixing, and thereby
sediment dynamics and residual transport. The collected data suggests that sediment-induced turbulence damping
weakens sediment transport in the flood direction (Bailey et al., in prep.).
- The cross-sectional data revealed pronounced transverse flows in the ENC despite its limited width (~500 m). The
collected data, with frame and ship-borne measurements on both sides of the channel provide information to
determine the lateral and longitudinal density gradients driving these complex flow patterns. It is recommended to
investigate these lateral flows in greater detail, including its role on residual sediment transport.
- Based on data only it is not yet feasible to exactly quantify the role of the lower Ems River on the turbidity in the
outer Ems estuary. The impact of sediment flushing from the lower Ems River and the ENC on turbidity in the outer
Ems estuary requires detailed further analysis of the data (for instance through decomposition of fluxes, as above)
in combination with numerical modelling (for which the EDoM data provides valuable calibration data).
- Regarding the exchange of sediments between the lower Ems River and the ENC, i.e. upstream transport versus
downstream flushing, the region of the highest along-channel sediment-induced density gradient downstream of the
fluid mud reach is critical in understanding the Ems system, but was not part of the EDoM survey. It is
recommended to complement the EDoM data set by conducting new measurements in this part of the channel,
upstream of Pogum and including the downstream part of the fluid mud reach.



## 6 Data availability

Most data collected during the EDoM field campaign is stored on the repository of 4TU: https://doi.org/10.4121/c.6056564.v3 (van Maren et al., 2022). Most data stored on the repository is averaged at 10-minute intervals; water sample, settling velocity and turbulence data is stored at different intervals. For more details on the data itself, or access to the original (non-averaged) data, contact authors responsible for collection of the data of interest (see Table 2 for an overview of the responsible institute per measurement location). We encourage use of the EDoM data by non-participants of the measurement campaign although

all use of the data should be communicated with the responsible surveyors.

## 7 Conclusions

With 8 ships and 10 moorings concurrently measuring water levels, flow velocities, salinity and turbidity, the EDoM dataset provides a unique dataset to obtain synoptic patterns of residual flow and sediment transport. The shipborne surveys additionally provide detailed data to investigate vertical mixing processes, while the mooring allow assessment of processes

operating at spring-neap tidal cycles. An integral analysis of these observations suggest that large-scale residual transport is remarkably similar during periods of high and low discharge, with sediment exchange being strongly influenced by a lateral circulation cell driving residual transport. Potentially, flow and sediment transport over the dam separating the Dollard Bay from the ENC is important for exchange flows, but this has not yet been corroborated by measurements. Vertical density-driven flows in the outer estuary are influenced by variations in river discharge, with a near-bed landward flow being most

pronounced in the days following a period with elevated river discharge. This is relevant for the large-scale landward sediment transport that exists in the outer estuary, although an asymmetry in the duration of slack water (with a longer duration of high water slack) appears to be more important. The study site is more turbid during winter conditions, when the Estuarine Turbidity Maximum is pushed seaward by river flow, resulting in more pronounced impact of suspended sediments on hydrodynamics. In terms of data analysis, this paper focussed on an integral analysis of all data and synoptic patterns of sediment transport and

residual flow. However, much more insight into transport and exchange mechanisms may be obtained through more detailed further analyses, and by combining the dataset with numerical models.

## Author contributions

DSM initiated the field campaign and contributed its technical organisation, assisted during the field observations, analysed data and wrote the manuscript. CM and JWM setup the technical aspects of the field campaign and organised the campaign

from the German and Dutch side (resp). DvK, JJ, JV and MB analysed the collected ADCP and CTD data. HS initiated the field campaign, and assisted during the field campaign and through data analysis. TG and KS collected and processed the 2019 SB_POG data, DMLH collected and analysed the LISST data. THB and MG collected and processed the data at SB_POG and CS_POG. TB, LR and VM collected and processed the turbulence data. AE provided the monitoring data collected by German



authorities. DP and CS coordinated the campaign from the German and Dutch side (resp.). PJTD was responsible for overall

coordination and logistics. All co-authors contributed to the manuscript.

**Competing interest**

The authors declare that they have no conflict of interest.

**Acknowledgements**

We would like to thank the Port of Emden to making their survey vessel Delphin available, and the Wasserstraßen- und Schifffahrtsamt WSA for providing their research vessel Friesland. We are very grateful to the crews of the Navicula (NIOZ), Asterias (RWS), Amasus (RWS), Delphin (PE / BAW), Friesland (WSA / BAW), Otzum (Oldenburg), Zephyr (Oldenburg), Scheurrak (RWS), Burchana (NLWKN) for their dedicated contributions.


**Literature**

Bailey, T., L. Ross, H. Schuttelaars, and D.S. van Maren: Implications of Sediment-Induced Stratification in a Meso-Macrotidal Estuary, manuscript in preparation.

Becker, M., Maushake, C., & Winter, C: Observations of mud-induced periodic stratification in a hyperturbid estuary.
Geophysical Research Letters, 45, 5461–5469. https://doi.org/10.1029/2018GL077966, 2018.

Burchard, H., H. M. Schuttelaars and D. K. Ralston (2017). "Sediment Trapping in Estuaries." Annual Review of Marine Science.

Chernetsky AS, Schuttelaars HM, Talke SA: The effect of tidal asymmetry and temporal settling lag on sediment trapping in tidal estuaries. Ocean Dyn 60(5):1219–1241, 2010.

Dijkstra, Y. M., H. M. Schuttelaars, and G. P. Schramkowski: A Regime Shift From Low to High Sediment Concentrations in a Tide-Dominated Estuary, Geophysical Research Letters, 46(8), 4338-4345, 2019a.

Dijkstra, Y. M., H. M. Schuttelaars, G. P. Schramkowski, and R. L. Brouwer: Modeling the transition to high sediment concentrations as a response to channel deepening in the Ems River Estuary, Journal of Geophysical Research: Oceans, 124(3), 1578-1594, 2019b.

Dyer, K. R.: Fine sediment particle transport in estuaries. In Physical processes in estuaries (pp. 295–310). Berlin, Heidelberg: Springer.https://doi.org/10.1007/978-3-642-73691-9_16, 1988.



Dyer, K.R.: Estuarine sediment transport and deposition. In: Pye, K. (Ed.), Sediment Transport and Depositional Processes. Blackwell Scientific Publications, Oxford, pp. 193–218, 1994.

Dyer, K.R., Christie, M.C., Feates, N., Fennessy, M.J., Pejrup, M., Vander Lee, W: An investigation into processes influencing the morphodynamics of an intertidal mudflat, the Dollard Estuary, The Netherlands: I. Hydrodynamics and suspended sediments. Estuar. Coast. Shelf Sci. 50, 607–625, 2000.

Efron, B., Gong, G.: A leisurely look at the bootstrap, the jackknife and cross-validation. The American Statistician, 37, 36-48, 1983.

Friedrichs, C. T.: Tidal flat morphodynamics: a synthesis. Treatise on Estuarine and Coastal Science: Sedimentology and Geology, 2011.

Huguenard, K., Bears, K., Lieberthal, B.: Intermittency in Estuarine Turbulence: A framework toward limiting bias in microstructure measurements, Journal of Atmospheric and Oceanic Technology, 36(10), 1917-1932, 2019.

de Jonge, V.N., Schuttelaars, H.M., van Beusekom, J.E.E., Talke, S.A., de Swart, H.E.: The influence of channel deepening on estuarine turbidity levels and dynamics, as exemplified by the Ems estuary. Estuarine, Coastal and Shelf Science 01/2014; DOI:10.1016/j.ecss.2013.12.030, 2014.

Jay, D.A., and Musiak, J.D.: Particle trapping in estuarine tidal flows. J. Geophys. Res. 99, 445-461, 1994.

Jensen, J., Frank, T., Mudersbach, C.: Dokumentation und Untersuchungen zur Begleitung der Beweissicherungsmessungen Emssperrwerk (Null-Messung). NLWKN report WBL 156 D, 102 p., 2002.

Lerczak, J. A., and W. R. Geyer: Modeling the lateral circulation in straight, stratified estuaries. J. Phys. Oceanogr., 34, 1410–1428, 2004.

Manning, A.J., and Dyer, K.R.: The use of optics for the in situ determination of flocculated mud characteristics. J. Opt. A: Pure Appl. Opt. 4 (2002) S71–S81, 2002.

Pein, J.U., E. V. Stanev, and Y. S. Zhang: The tidal asymmetries and residual flows in Ems Estuary. Ocean Dynamics 64.12, p. 1719-1741, 2014.

Postma, H.: Sediment transport and sedimentation in the marine environment. Estuaries, Special Publication 83, 158–179, 1967.

Ross, L., Huguenard, K. D., Sottolichio, A.: Intratidal and fortnightly variability of vertical mixing in a macrotidal estuary: The Gironde. Journal of Geophysical Research: Oceans, 124, p. 2641-2659. https://doi.org/10.1029/2018JC014456, 2019.

Schulz, K., Burchard, H., Mohrholz, V., Holtermann, P., Schuttelaars, H. M., Becker, M., ... & Gerkema, T.: Intratidal and spatial variability over a slope in the Ems estuary: Robust along-channel SPM transport versus episodic events. Estuarine, Coastal and Shelf Science, 243, 106902, 2020.

Schuttelaars, H.M., de Jonge, V.N., Chernetsky, A.: Improving the predictive power when modelling physical effects of human interventions in estuarine systems, Ocean & Coastal Management, doi:10.1016/j.ocecoaman.2012.05.009, 2013.

Simpson, J.H., Brown, J., Matthews, J.P., Allen, G.: Tidal straining, density currents and stirring in the control of estuarine stratification. Estuaries 13 (2), 125–132, 1990.





Spingat, F. and Oumeraci, H.: Schwebstoffdynamik in der Trubungszone des Ems-Astuars. Die Kuste, 62, pp. 159–219, 2000.

Talke, S.A., H.E. de Swart, and H.M. Schuttelaars: Feedback between residual circulations and sediment distribution in highly turbid estuaries: an analytical model. Continental Shelf Research 29: 119–135. doi:10.1016/j.csr.2007.09.002, 2009.

Van de Kreeke J, Robaczewska K: Tide-induced residual transport of coarse sediment: Application to the Ems estuary. Neth
J. Sea Res 31(3):209–220, 1993.

van Maren, D.S., van Kessel, T., Cronin, K., Sittoni, L: The impact of channel deepening and dredging on estuarine sediment concentration. Continental Shelf Research 95, p. 1-14 http://dx.doi.org/10.1016/j.csr.2014.12.010, 2015a

van Maren, D.S., Winterwerp, J.C., Vroom, J.: Fine sediment transport into the hyperturbid lower Ems River: the role of channel deepening and sediment-induced drag reduction, Ocean Dynamics, DOI 10.1007/s10236-015-0821-2, 2015n.

van Maren, D.S.; Mol, J.W. (Jan Willem); Maushake, Christian; Gerkema, Theo; Vroom, Julia; van Keulen, Daan; et al.: The Ems-Dollard Measurement (EDoM) campaign 2018 - 2019. 4TU.ResearchData. Collection, 2022.

Vroom, J., de Vries, B., Dankers, P.J.T., and van Maren, D.S.: Cumulatieve effecten baggeren en verspreiden op habitattype H1130 in het Eems estuarium. Unpublished Deltares report, 2022.

Winterwerp, J. C., Manning, A. J., Martens, C., de Mulder, T., & Vanlede, J.: A heuristic formula for turbulence-induced
flocculation of cohesive sediment. Estuarine, Coastal and Shelf Science, 68(1), 195-207. DOI: 10.1016/j.ecss.2006.02.003, 2006.

Winterwerp, J.C.: Fine sediment transport by tidal asymmetry in the high-concentrated Ems River: indications for a regime shift in response to channel deepening. Ocean Dynamics 61:203-215, 2011.

Winterwerp JC, Wang ZB, van Braeckel A, van Holland G, Kösters F: Man-induced regime shifts in small estuaries – I: a
comparison of rivers. Ocean Dyn 63 (11–12):1293–1306, 2013.

Winterwerp, J. C., Vroom, J., Wang, Z. B., Krebs, M., Hendriks, E. C., van Maren, D. S., Schrottke, K, Borgsmuller, C., and Schöl, A.: SPM response to tide and river flow in the hyper-turbid Ems River. Ocean Dynamics, 67(5), 559-583, 2017.

## Figures

![Figure 1 map]

**Figure 1 Location of all observation stations during the EDoM campaigns. Top panel: Main water bodies referred to in this document (blue text) with long term mooring stations (black text). The green station measures wave height, the blue stations measure SSC and salinity (and for Knock and Terborg also water levels); the black stations (Borkum and Papenburg) are the end of the longitudinal transect sailed during the 13-hrs surveys. The diamond denotes the weir at Herbrum and for which the discharge measured at Versen (further landward) is used as river discharge. Lower panel: detail with stations occupied during the EDoM**
**campaigns (see Table 1 for explanation of abbreviations).**



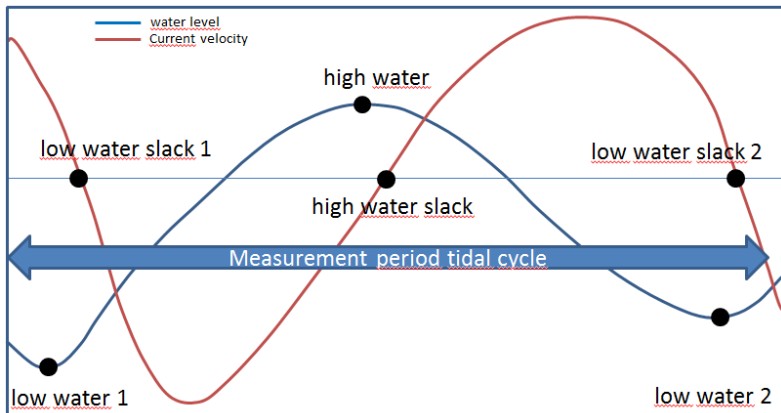

**Figure 2** **Definition of the measurement period of the 13-hrs surveys.**



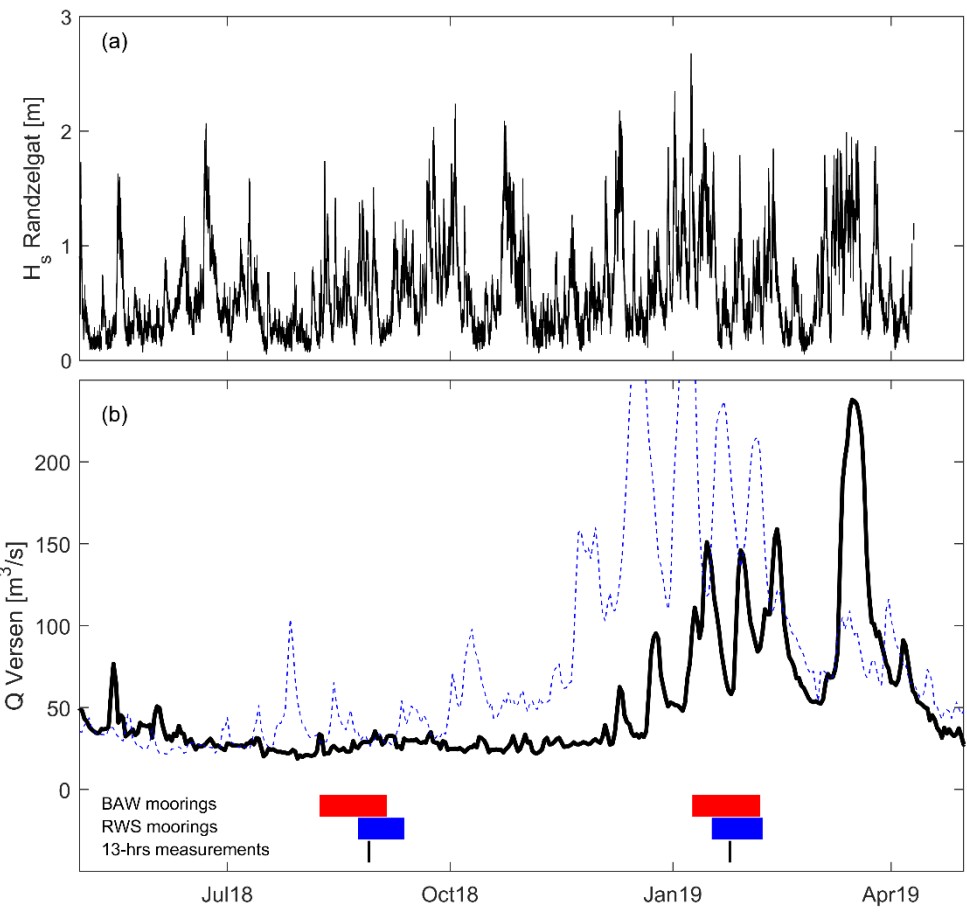

**Figure 3 Wave height H$_s$ measured at station Randzelgat Noord (a) and discharge of the Ems River at Versen (b) from 1 May 2018 to 1 May 2019, with deployment dates of BAW frames, RWS frames, and the 13-hrs measurements added to (b). The dashed blue discharge in (b) is the discharge of the period 1 May 2017 to 1 May 2018.**



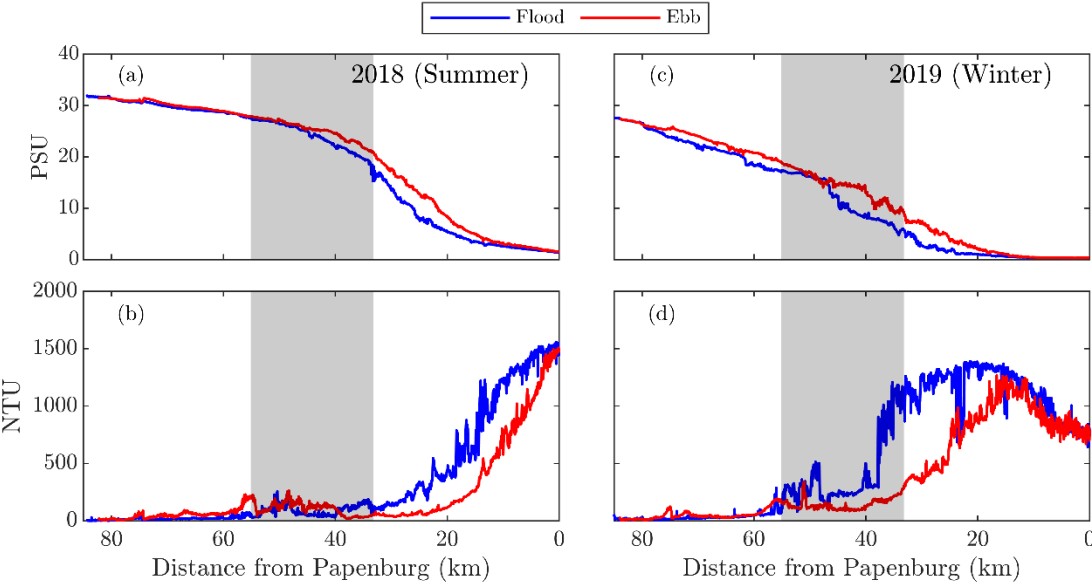


**Figure 4** Longitudinal near-surface salinity (a,c, in PSU) and turbidity (b, d, in NTU) distribution observed in 2018 (a,c) and 2019 (b,d), during the flood (blue) and during the ebb (red) cruise. The survey starts at km 85 (Borkum) at the beginning of flood and reaches Papenburg around the transition from flood to ebb after which it sails back for 6 hours with the ebbing tide. The grey-shades area denotes the focus area of the EDoM campaign. Observations were made with a near-surface sensor towed by the ship, 650 and therefore no near-bed observations are available.





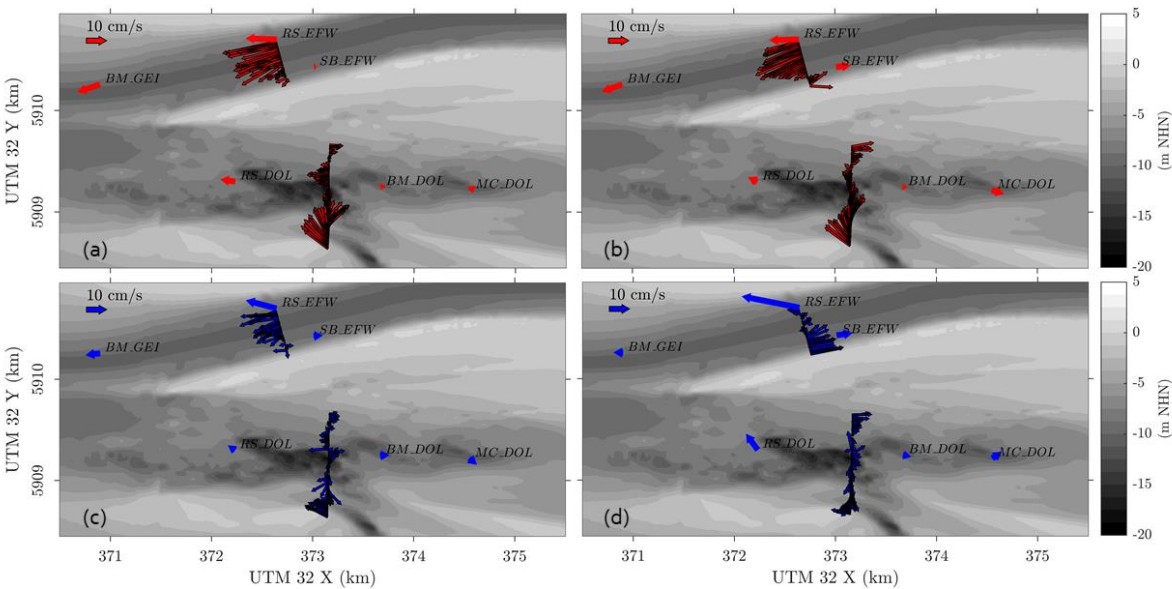


**Figure 5** **Residual flow velocity near-surface (top) and near-bed (below) over the 13-hr measurement period on 28 August 2018 (a, b) and 24 January 2019 (c,d), including moorings (BM_GEI, RS_EFW, RS_DOL, BM_DOL, MC_DOL), shipborne stationary observations (SB_EFW) and the transects in the Fairway to Emden (CS_EFW) and in the Dollard (CS_DOL).**






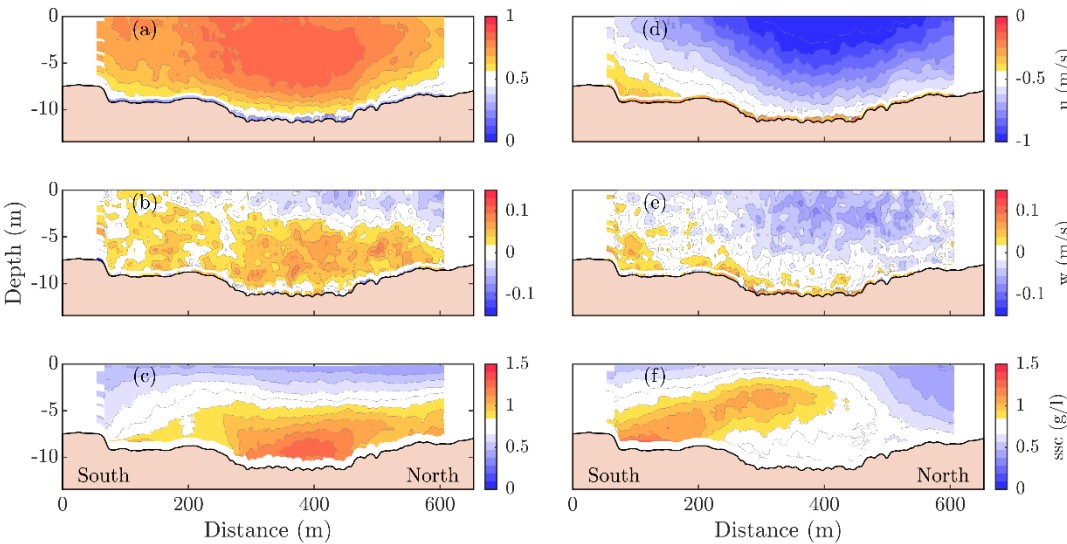

**Figure 6 Velocity and SSC measured at CS_EFW cross-section in 2019 averaged over the flood (left panels a, b, and c) and ebb (right panels; d, e, f). Panel a and d: measured along-channel current velocities; panels b and d: cross-channel current velocities (northward positive) and panel e and f: SSC based on ADCP backscatter conversion.**
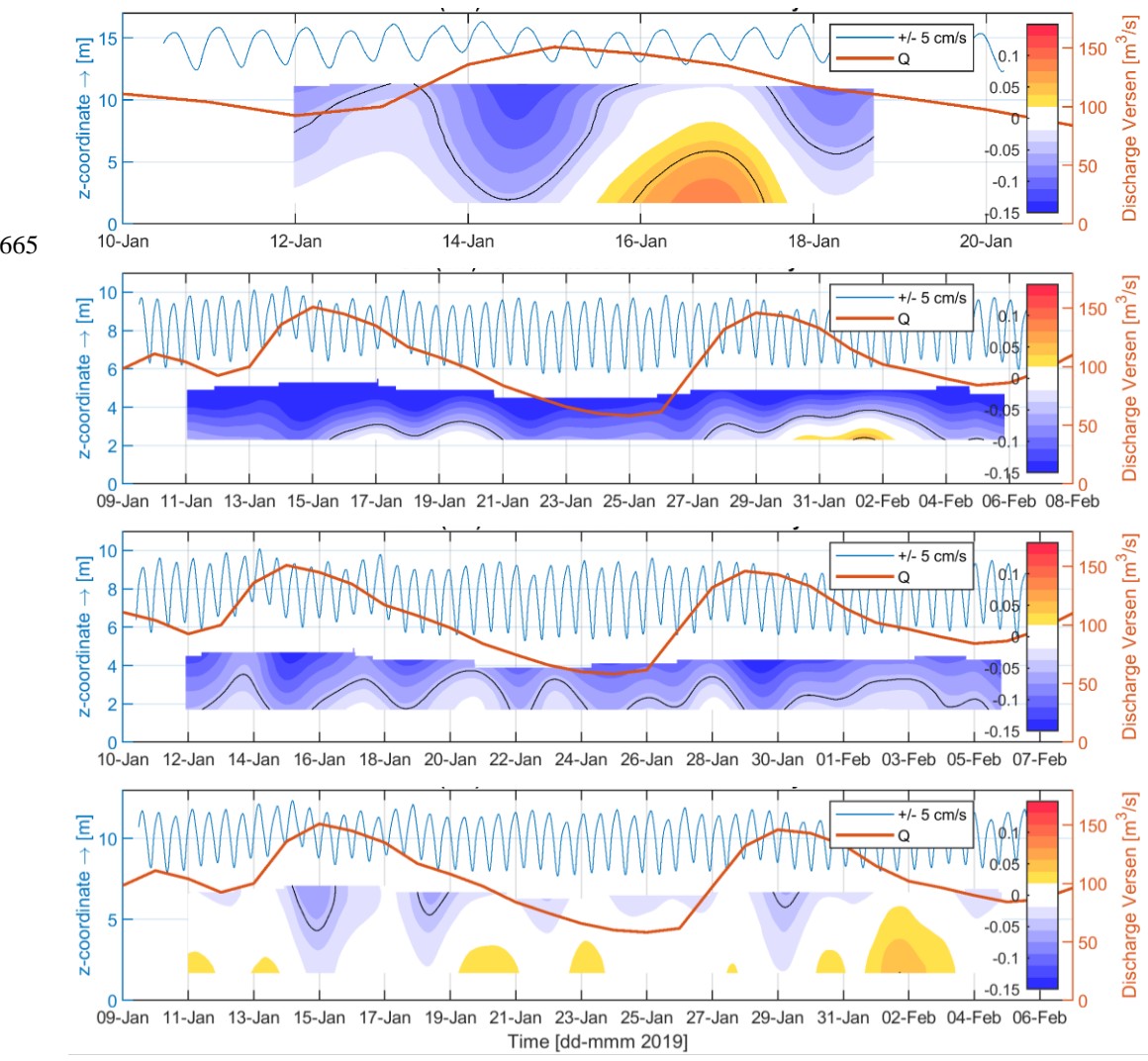

**Figure 7 Godin-filtered residual longitudinal flows (landwards positive) measured with the bottom-mounts (BM) at stations GAT (panel 1), GEI (panel 2), EFW (panel 3) and DOL (panel 4) in 2019. A Godin low-pass filter removes tidal flow velocities from the observation, showing temporal variations of the average flow velocity.**

**Figure 8 Residual sediment transports [kg/m²/s] computed at all observation stations with a length exceeding the duration of a spring-neap cycle, in 2018 (a) and 2019 (b) for which velocity and SSC observations are available. At MC stations, velocities at 3 positions in the vertical are multiplied with SSC at the same position. At other locations velocity profiles, as collected by an ADCP are multiplied with near bed SSC.**





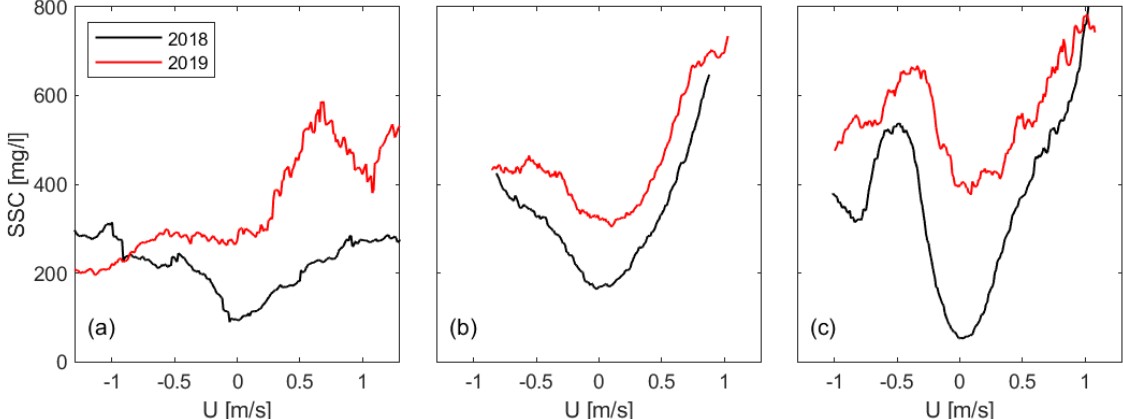

**Figure 9 Sediment concentration averaged per eastward flow velocity U (flood currents positive) over a spring-neap tidal cycle for the moored observations in the Dollard: RS_DOL (a), MC_DOL (b), and RM_DOL (c).**





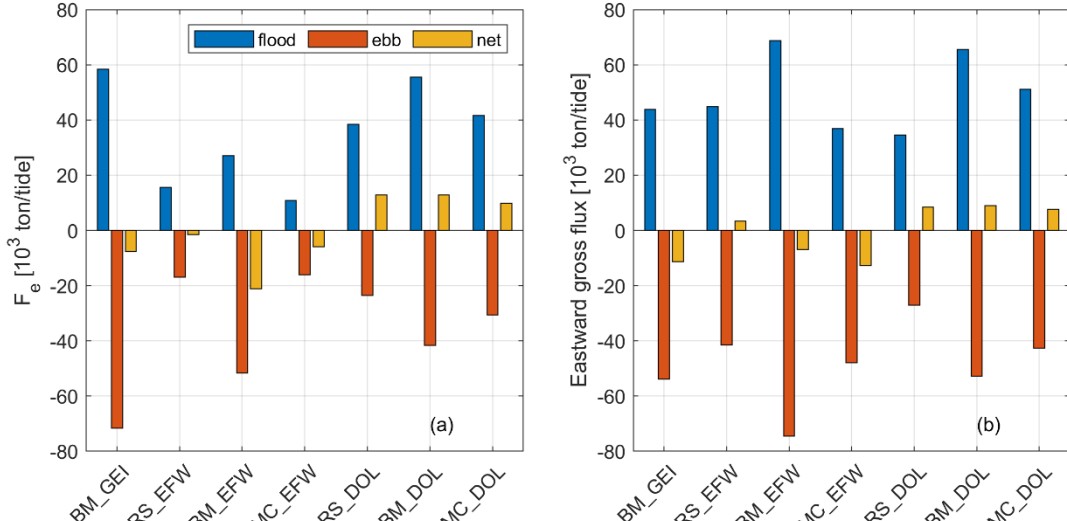

**Figure 10** Gross and net flux $F_e$ (eastward positive), computed from one spring-neap cycle of observations at the long term moorings in 2018 (a) and 2019 (b).



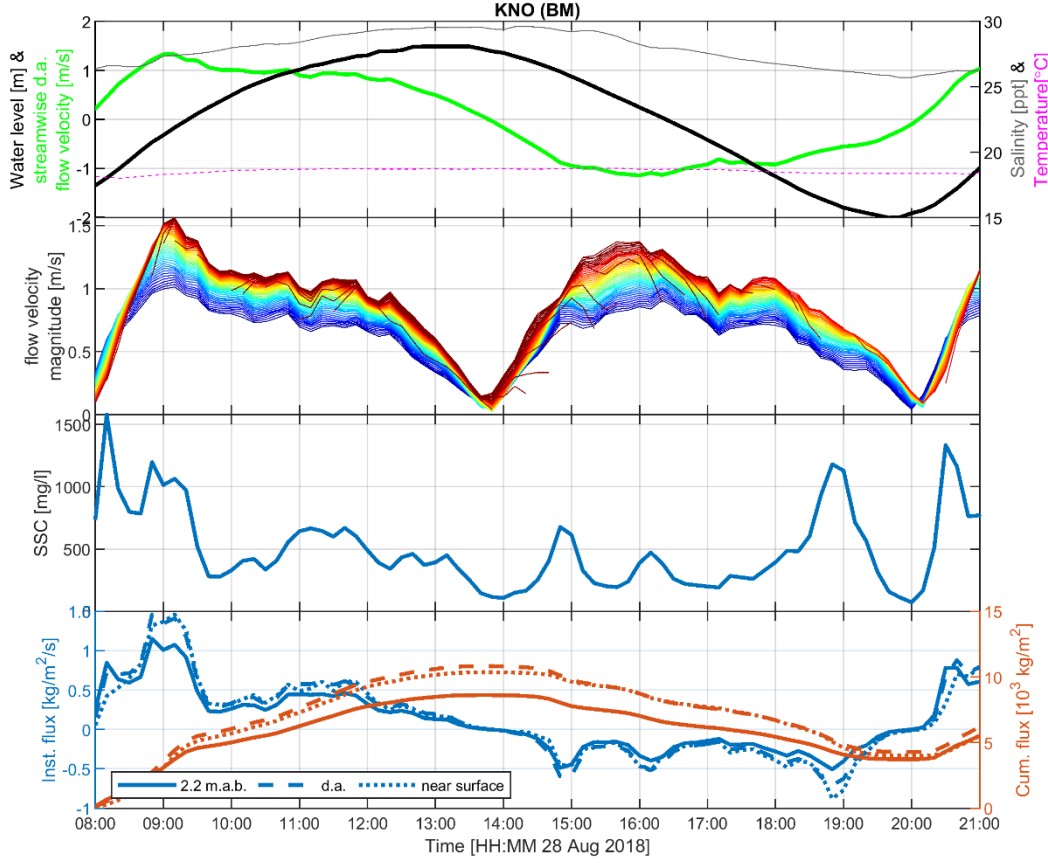

**Figure 11** Tidal cycle observations at Knock (BM_KNO), showing the water level (black), salinity (grey), depth-averaged flow velocity (green) and temperature (pink, top panel), the depth-varying flow velocity (red near the surface, blue near the bed, second panel), the sediment concentration near the bed (third panel), and instantaneous and cumulative sediment flux (landwards positive; lower panel).

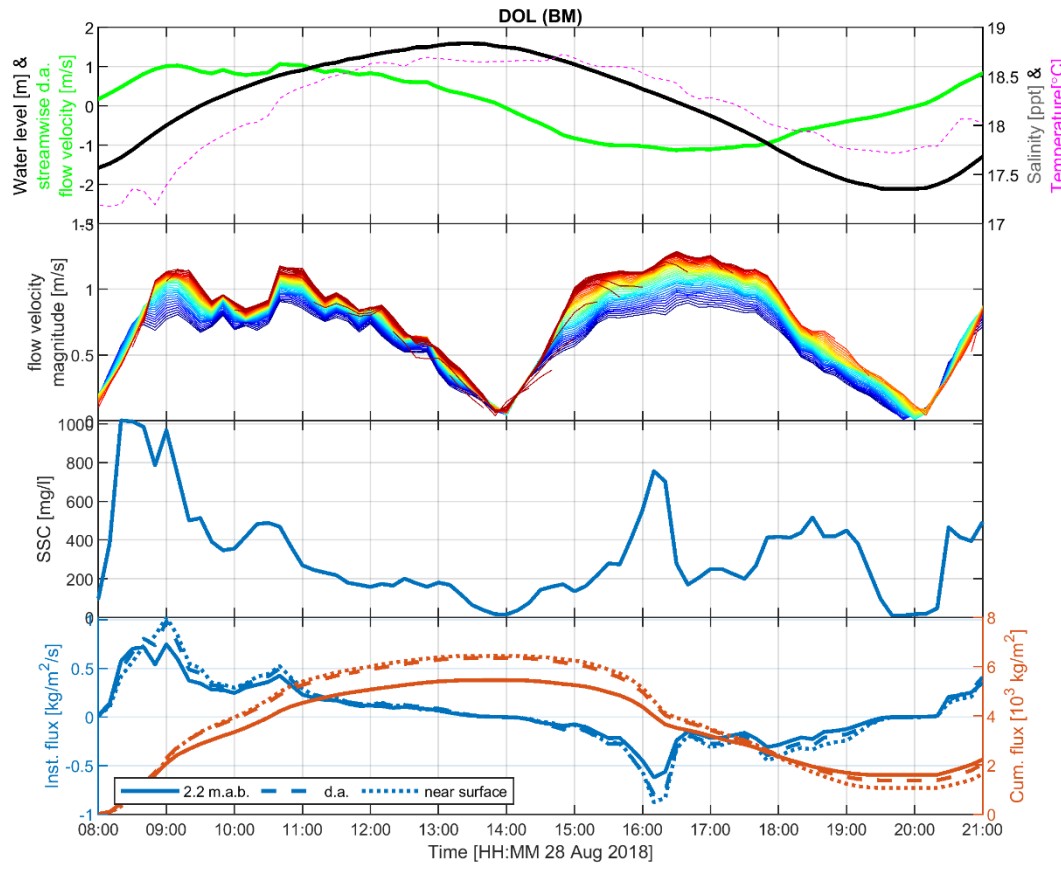

**Figure 12** Tidal cycle observations at mouth of the Dollard (BM_DOL), showing the water level (black), salinity (grey),
depth-averaged flow velocity (green) and temperature (pink, top panel), the depth-varying flow velocity (red near the surface, blue
near the bed, second panel), the sediment concentration near the bed (third panel), and instantaneous and cumulative sediment flux
(landwards positive; lower panel).



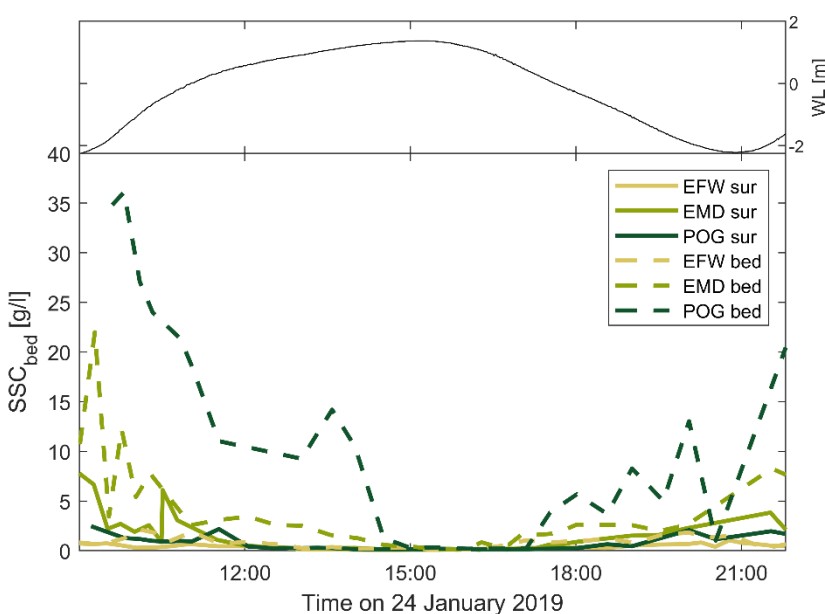

**Figure 13**              **SSC based on water samples collected in the Fairway to Emden (station SB_EFW and SB_EMD) and the lower Ems River (SB_POG), on 24 January 2019 near the bed (top) and near the surface (bottom).**



**Tables**

**Table 1 Explanation of abbreviations for survey type and location. At one specific location multiple measurement types may be collected (e.g. SB_EFW, RS_EFW, BM_EFW, and CS_EFW)**

| Measurement type | | Location | |
|---|---|---|---|
| **CS_** | Cross-section | GAT | Gatjebogen |
| **SB_** | Stationary Boat | KNO | Knock |
| **BM_** | Bottom Mount | DOL | Dollard |
| **MC_** | Mooring Chain | EFW | Fairway to Emden |
| **RS_** | RWS bottom frame | EMD | Emden |
| | | POG | Pogum |





**Table 2  Summary of executed measurements per location: name of observation station, Institute in charge of survey vessel, executed measurements, and measurement period. The measurements executed at Pogum (CS_POG and SB_POG, both in italics) suffered from high SSC values, corrupting the OBS and ADCP data. OBS and ADCP data should be carefully processed and interpreted, and therefore only water sample and temperature and salinity profile results are presented as part of this dataset. Observation station BM_KNO suffered from mechanical failure in 2019, and BM_GAT measured for only 10 days in 2019.**

| Observation station | Institute | Measurements | Period |
|---|---|---|---|
| **BM_GAT** | BAW | Velocity profile; salinity, temperature, turbidity at 0.5 m.a.b. | 8 August – 5 September 2018<br>10 January 2019 – 20 January 2019 |
| **BM_KNO** | BAW | Velocity profile; salinity, temperature, turbidity at 0.5 m.a.b. | 9 August – 2 September 2018<br>*(failure in 2019)* |
| **BM_GEI** | BAW | Velocity profile; salinity, temperature, turbidity at 0.5 m.a.b. | 9 August – 5 September 2018<br>9 January 2019 – 7 February 2019 |
| **BM_DOL** | BAW | Velocity profile; salinity, temperature, turbidity at 0.5 m.a.b. | 8 August – 5 September 2018<br>9 January 2019 – 6 February 2019 |
| **BM_EFW** | BAW | Velocity profile; salinity, temperature, turbidity at 0.5 m.a.b. | 9 August – 5 September 2018<br>10 January 2019 – 7 February 2019 |
| **MC_KNO** | BAW | Velocity, salinity, temperature, and turbidity at 1.5, 3.5, and 7.8 m.a.b. | 6 August – 3 September 2018<br>8 January 2019 – 5 February 2019 |
| **MC_DOL** | BAW | Velocity, salinity, temperature, and turbidity at 1.5, 3.5, and 7.9 m.a.b. | 6 August – 3 September 2018<br>8 January 2019 – 5 February 2019 |
| **MC_EFW** | BAW | Velocity, salinity, temperature, and turbidity at 1.5, 3.5, and 7.7 m.a.b. | 6 August – 3 September 2018<br>8 January 2019 – 5 February 2019 |
| **RS_DOL** | RWS | Velocity profile (also near-bed); salinity, temperature, turbidity at 0.2, 0.3, 0.5 and 0.8 m.a.b. | 24 August – 12 September 2018<br>16 January 2019 – 7 February 2019 |
| **RS_EFW** | RWS | Velocity profile (also near-bed); salinity, temperature, turbidity at 0.2, 0.3, 0.5 and 0.8 m.a.b. | 24 August – 12 September 2018<br>16 January 2019 – 7 February 2019 |
| **SB_KNO** | NIOZ (2018) / RWS (2019) | Profiles of salinity, temperature, turbidity, velocity, settling velocity (LISST200x) and turbulence (settling velocity and turbulence only in 2018); water samples near-surface, near-bed, and in the middle | 28 August 2018 and 24 January 2019 |
| **SB_EFW** | BAW | Profiles of salinity, temperature, turbidity, velocity; water samples near-surface, near-bed, and in the middle | 28 August 2018 and 24 January 2019 |
| **SB_EMD** | RWS | Profiles of salinity, temperature, turbidity, velocity, settling velocity from camera; water samples near-surface, near-bed, and in the middle | 28 August 2018 and 24 January 2019 |
| **SB_POG** | Oldenburg | Profiles of salinity, temperature, oxygen, turbidity, water samples near-surface and near-bed | 28 August 2018 and 24 January 2019 |
| **CS_DOL** | RWS | Profiles of flow velocity and echo intensity | 28 August 2018 and 24 January 2019 |





| CS_EFW | BAW | Profiles of flow velocity and echo intensity | 28 August 2018 and 24 January 2019 |
|---|---|---|---|
| CS_POG | Oldenburg | Profiles of flow velocity and echo intensity, salinity, temperature (2019) and additional turbidity, chlorophyll and CDOM (2018) at 0.7 m | 28 August 2018 and 24 January 2019 |
| Longitudinal | NLWKN | Near-surface salinity, temperature and turbidity; profiles of echo intensity and velocity | 28 August 2018 and 24 January 2019 |
| Permanent | NLWKN | Water levels, SSC/ salinity/oxygen/temperature | July 2017 – June 2019 |
