# Peer review of "Synoptic observations of sediment transport and exchange mechanisms in the turbid Ems estuary: the EDoM campaign"

_Earth System Science Data, 2022_

## Referee Comment (RC2)

Dear authors and editor,

The paper "Synoptic observations of sediment transport and exchange mechanisms in the turbid Ems estuary: the EDoM campaign" provides the data and preliminary analyses of an extensive field campaign in the Ems, carried out by an excellent team. There is no doubt that this is an impressive data set that needs to be published. The Journal ESSD Ocean would be a suitable journal.

My concern is a bit more in how the manuscript is written now. My three main points are:

1. The manuscript is written from the viewpoint of what the authors wanted to do with the data set and what they would like to do with it. This makes sense for a data report. By publishing in ESSD, you would however also invite other researchers to work with the data. Some additional paragraph(s) in the introduction and Section 5 would help then.

2. Although the data provides a wealth of interpretation options that are not fully explored yet, the present manuscript is already quite extensive in interpretation. The aim of the journal ESSD mentions: "*Any interpretation of data is outside the scope of regular articles.*", see https://www.earth-system-science-data.net/about/aims_and_scope.html. Some interpretation will of course help the reader understanding the data, but the authors could reconsider whether all descriptions/interpretations are really needed. Some could better be used in a follow-up paper where the focus is on the interpretation.

3. Processed data is provided, without the raw data or high-resolution data. Some data also seems to be missing, while other data is averaged over 10 minutes. This makes it difficult/impossible to check the data or using it for different types of analysis. This limits the use of the data. For publication, I would highly recommend to upload the full data sets, including high-frequency processed (despiked) results and raw data.

Suggestions below are mainly along the same lines.

Section 1:
The three indicated research questions (line 78 onwards) are very specifically framed towards the Ems. This framing was of course needed to justify the campaign. These measurements could however also help to understand other systems where such campaigns were not performed. The introduction could put the campaign in a wider perspective, to attract potential users.

Furthermore, the questions are also not that clear yet. For example: Line 78 onwards: *The first is that while transport mechanisms in the lower Ems River has been studied in great detail (see references above) less … sediments landward (van Maren et al., 2015a).* From this description, it is hard to distill the exact research question. Rewriting in shorter sentences might help. Line 422 already provides a clearer question: "*One of the motivations for the EDoM campaign was to understand the mechanisms leading to landward transport in the ENC.*".

**Section 2**

This section discusses the design of the experiment. Subsections 2.1-2.3 do however mainly discuss processes and have some overlap with the introduction. It is not (always) clear why these sections are really needed for the design of the campaign (2.4). Subsection 2.4 is quite brief. Some more considerations on why these exact locations were chosen would help the reader. Why not a larger/smaller area of interest? Why this setting of moorings, frames and boats?

**Section 3**

This section discusses the instrumentation. A clear overview is given. Also here, a few considerations would help. For example, why are the instruments on the mooring chains positioned at 1.5, 3.5 and ~7.8 m above the bed?

Section 3.2 discusses the data processing. Outliers were removed, but how? Is the raw data stored as well (in case outliers might have been interesting data)? Can you indicate with a few lines how calibration of the OBSes was done and whether the calibration curves are in the data set?

Line 211: "*However, with the high concentrations in the Ems estuary acoustic and optical instruments become progressively less reliable, making accurate water sampling an important source of data. In order to minimise the potential impact of methodological errors all water samples were therefore analysed in the same laboratory.*". This directly raises the question how (un)reliable the data are. In view of the use of the data by others, a reflection of the authors on this would help. In this paper, I would prefer that above an extensive interpretation of the data.

**Section 4**

This section starts directly with interpretation of the data. Would it be possible to include a section on how well the instruments worked? For example, did OBSes reach their upper limit? Any biofouling? How (un)reliable were the acoustic and optic instruments in the high concentrations? Is a comparison made between samples and acoustic/optic instruments? What were the findings? See also line 211.

Section 4.3 is not balanced with the previous subsections, where the data is elaborated in more detail. Some of the findings have been published already apparently. It was also not directly clear where to find the floc data.

Section 5

Section 5.1 gives a clear overview of the findings. As indicated above, Section 5.2 mainly discusses the plans of the project team. Some of the mentioned plans for future research likely started already, as the data set is there for a couple of years now. This section could be modified to make the paper more attractive for other researchers. How could and why should researchers from outside the project team use the data set? Which type of data could be used for comparisons between estuaries for example? Which data can be used for intercomparison of instruments? What kind of things could not be analyzed by the research team, because a lack of experience?

Section 6

The data availability section is not clear enough. It states that "most data" is stored on the repository. This directly raises the question why not all data is stored there, and which data is missing and why. Data has been averaged to 10-minute intervals, reducing the possibilities for analyses outside the project team. The statement in line 219: "*We encourage use of the EDoM data by non-participants of the measurement campaign although all use of the data should be communicated with the responsible surveyors.*" is a bit weird. I would recommend to open the data set completely, without such restrictions.

The data on the repository is divided over different directories. It is now a bit difficult to find the observation stations from Table 2 in the repository. Adding the directories in Table 2 might help?

---

## Author Comment (AC1)

This paper presents an overview of a complex field campaign (multiple vessels, multiple moorings, two seasons) conducted in the lower Ems River, outer Ems Estuary, and dredged navigation channel which connects them. The motivating question is what mechanisms drive the high sediment concentrations and high rates of sedimentation (necessitating annual maintenance dredging) in the channel. This topic has been investigated before in this and other estuarine-tidal river systems, but the authors suggests that mechanisms traditionally cited (tidal asymmetry, residual circulation, etc.) seem too weak in themselves to explain the high SSC/trapping here. Part of the summary notes that exchange between the Dollard and channel may provide a temporary source/sink coupling (which fluctuates with river discharge) which causes the unusually high trapping. The authors note that decomposition of fluxes into residual, tidal pumping, and other terms may help fully explain the dynamics. In similar papers these calculations are usually included already, but this paper has been written to describe the dataset (and submitted accordingly to a journal dedicated to that purpose) and it is suggested that future papers will explore the dynamics in detail.

Specific comments include:

1. Line 61 – subject/verb agreement

   *I am afraid we do not understand what is wrong with this sentence. Please make more clear what is wrong (if considered important)*

2. Line 81 – grammar/typo – "even though model result that"

   *Corrected (result ➔ reveal)*

3. Line 85 – typo "ion"

   *Corrected*

4. Line 89-90 – As written, this sentence is somewhat confusing (based on the sentences which precede it)

   *The reviewer is correct that the wording was unclear. We changed*

   *'This points to the presence of convergence of suspended sediment transport, which in turn conflicts with the large dredging volumes in the lower Ems River suggesting up-estuary transport. '*

   *Into*

   *'The only way to then explain the high siltation rates in such a dynamic area is is points to the presence ofstrong convergence of suspended sediment transport. However, strong convergence of sediment transport in the ENC conflicts with the large up-estuary transport discussed above.'*

5. Throughout the paper – estuary should be capitalized in Ems Estuary, just as in Ems River

   *Corrected*

6. 112 – tons (plural)

   *Corrected*

7. Line 131 – this section discusses sediment transfer between the outer Ems Estuary and lower Ems River, and notes that strong flood dominance may promote trapping – but in line 81 the authors note that tides in the ENC (the navigation channel connecting those two regions) the tides are "asymmetric with higher ebb flow velocities." Is there a spatial switch between flood dominant flows in the outer estuary to ebb dominant flows in the ENC to flood dominant flows in the lower Ems River? Please clarify.

   *The confusion arising from the fact that we did not clearly mention earlier that the lower Ems River is very flood-dominant – the ENC is the anomaly here. Therefore the following sentence one section earlier*

   *'Approximately 1 to 1.5 million tons are annually extracted from the lower Ems River by dredging'*

   *Has been replaced by*

   *'The lower Ems River is strongly flood-dominant with a short period of high flood flow velocities (and a long period of weaker ebb flow velocities). The resulting trapping of sediments results in dredging requirements of approximately 1 to 1.5 million tons which are subsequently disposed on land'*

8. Line 195 – correct "turbiditymeter sensor"

   *Corrected (meter removed)*

9. 218 – is it possible that the pipette method biases the results toward smaller flocs? (Or did the diameters agree well with what the LISST reported?)

   *The subsampling does not influence the floc distribution. To better explain this we have provided more details on the subsampling by extending the section*

   *'Additional instrumentation was deployed onboard the stationary boats at Emden (SB_EMD) and Knock (SB_KNO) to measure turbulence and sediment settling properties. At SB_EMD, hourly water samples were taken with a Niskin bottle close to the bed and close to the water surface. A subsample taken with a pipette is inserted into a still and clear water settling column operated onboard, in which the water-sediment mixture settles from suspension. This settling is monitored with a high-resolution video camera. Postprocessing of the camera data reveals the size, shape, and settling velocity of all particles registered with the camera. '*

   *Into*

   *'Additional instrumentation was deployed onboard the stationary boats at Emden (SB_EMD) and Knock (SB_KNO) to measure turbulence and sediment settling properties. At SB_EMD, hourly water samples were taken with a Niskin bottle close to the bed and close to the water surface. A subsample taken with a pipette (with an orifice approximately 6-7mm in diameter, so large enough to not restrict large macroflocs passing through into the settling column) is inserted into a still and clear water*

*settling column (with the same temperature and salinity as the in situ fluid) operated onboard, in which the water-sediment mixture settles from suspension. This extraction technique has been successfully utilized in numerous recent laboratory flocculation studies (e.g. Mory et al., 2002; Gratiot and Manning, 2004; Graham and Manning, 2004; Mietta et al., 2009) and creates minimal floc disruption during acquisition transfer to the column. Settling is monitored with a high-resolution video camera, and postprocessing of the camera data reveals the size, shape, and settling velocity of all particles registered with the camera. '*

10. Line 235-237 – the wording is a bit awkward here

    *We agree. The sentence*

    *'The cross-sectional profiles were only deploying ADCPs except for CS_POG which towed a FerryBox (2018) or CTD (2019). Transects were continuously sailing back and forth over a GPS-steered track to cover the mouth of the Dollard (CS_DOL), the Emden fairway (CS_EFW) and the lower Ems River (CS_POG). '*

    *Is changed into*

    *'Only flow velocities (measured with ADCPs) were measured at the cross-sectional profiles CS_DOL and CS_EFW, whereas salinity and temperature were additionally measured at CS_POG (using a towed FerryBox in 2018) and a CTD in 2019).'*

11. Line 237-238 – did stratification impact the ADCP calibration? (I.e., by additional distortion of the sound signal which is difficult to correct?)

    *We used salinity profiles measured at nearby stations to account for that. This is now clarified by extending the sentence*

    *'The backscatter of the ADCP was calibrated to SSC using water samples collected at nearby stationary boats'*

    *Into*

    *'The backscatter of the ADCP was calibrated to SSC using water samples and CTD profiles collected at nearby stationary boats. The CTD profiles were used to compute backscatter attenuation by salinity and temperature, thereby accounting for stratification effects. The residual backscatter was calibrated to SSC using the water samples. '*

12. Line 425 – interesting result

    *Thanks*

13. Line 449 – sediment concentrations rather than sediment dynamics?

    *Indeed, corrected*

---

## Author Comment (AC2)

Dear authors and editor,

The paper "Synoptic observations of sediment transport and exchange mechanisms in the turbid Ems estuary: the EDoM campaign" provides the data and preliminary analyses of an extensive field campaign in the Ems, carried out by an excellent team. There is no doubt that this is an impressive data set that needs to be published. The Journal ESSD Ocean would be a suitable journal.

*Great, thanks.*

My concern is a bit more in how the manuscript is written now. My three main points are:

1. The manuscript is written from the viewpoint of what the authors wanted to do with the data set and what they would like to do with it. This makes sense for a data report. By publishing in ESSD, you would however also invite other researchers to work with the data. Some additional paragraph(s) in the introduction and Section 5 would help then.

*We do encourage other people to work with the data, and therefore all suggestions (as below) to make the dataset more accessible to other potential users is greatly appreciated and taken into account. As should become clear in our response to the more detailed comments below we have (1) stressed the generic problem more clearly (section 1), (2) better explained instrumental and methodological aspects (section 3), (3) provided recommendation on how to use the data for generic (not case-specific) research (section 5), and (4) explicitly invite external researchers to work with the data (section 5, 6). We have also provided more information on postprocessing of the data (section 3) and quality of the data (new section 4.1).*

2. Although the data provides a wealth of interpretation options that are not fully explored yet, the present manuscript is already quite extensive in interpretation. The aim of the journal ESSD mentions: "Any interpretation of data is outside the scope of regular articles.", see https://www.earth-system-science-data.net/about/aims_and_scope.html. Some interpretation will of course help the reader understanding the data, but the authors could reconsider whether all descriptions/interpretations are really needed. Some could better be used in a follow-up paper where the focus is on the interpretation.

*The reason why we have added interpretation is twofold. First, we need to present examples of the data as part of the data description. We can randomly show some graphs how the data looks like, but we have now presented our graphs along a storyline that we believe is much more interesting to the reader. The various comments of the reviewer did reveal that the framing of our work needed to improve – we have more clearly explained how the measurement campaign follows from the research questions (section 2), and how these research questions also have a more generic nature (section 1). Secondly, and most importantly, we hope our storyline inspires readers to further explore our dataset. Encouraging other users to work with the data is something the reviewer is missing, so apparently we failed at sufficiently conveying that message (see also reply above and more details below).*

3. Processed data is provided, without the raw data or high-resolution data. Some data also seems to be missing, while other data is averaged over 10 minutes. This makes it difficult/impossible to check the data or using it for different types of analysis. This limits the use of the data. For

publication, I would highly recommend to upload the full data sets, including high-frequency processed (despiked) results and raw data.

*Collecting such a large dataset with such a large amount of contributors requires agreements on how to work with and exchange data. One of the agreements we made (also prior to the campaign) is that we will share all 10-minute averaged data, and we will contact individual data collectors for more detailed data. That is a way of working we cannot reverse at this moment for a variety of reasons. This is now better explained (section 6).*

Suggestions below are mainly along the same lines.

**Section 1:**

The three indicated research questions (line 78 onwards) are very specifically framed towards the Ems. This framing was of course needed to justify the campaign. These measurements could however also help to understand other systems where such campaigns were not performed. The introduction could put the campaign in a wider perspective, to attract potential users.

*That is a very good suggestion. We believe such an introduction belongs more at the beginning of the introduction, where we changed the first sentence from*

*'The Ems estuary, located on the Dutch-German border, is heavily modified by human activities. '*

*Into*

*'Many estuaries worldwide, but particularly in Western Europe, have been deepened in the past decades to centuries, allowing ship access to inland ports. Both deepening and reclamation of intertidal areas have led to an increasing tidal range and salt intrusion, with tides penetrating increasingly deeper up-estuary. Hydrodynamics strongly control estuarine sediment dynamics (Burchard et al., 2018) and therefore the various human interventions have generally resulted in progressively higher turbidity levels (Winterwerp et al., 2013). Examples of heavily urbanized systems in which sediment dynamics have been modified by human interventions include the estuaries of the Elbe (Kerner et al., 2007, Winterwerp et al., 2013), the Weser (Schrottke et al., 2006), the Loire (Walther et al., 2012; Winterwerp et al., 2013), the Scheldt (Dijkstra et al., 2019c; Winterwerp et al., 2013) and the Yangtze Estuary (Zhu et al., 2021). The Ems estuaryEms Estuary, located on the Dutch-German border, is also heavily modified by human activities and is possibly the most thoroughly investigated system in terms of the relation between human activities and changes in turbidity. '*

*We believe this modification illustrates the wider potential of our work.*

Furthermore, the questions are also not that clear yet. For example: Line 78 onwards: The first is that while transport mechanisms in the lower Ems River has been studied in great detail (see references above) less … sediments landward (van Maren et al., 2015a). From this description, it is hard to distill the exact research question. Rewriting in shorter sentences might help. Line 422 already provides a clearer question: "One of the motivations for the EDoM campaign was to understand the mechanisms leading to landward transport in the ENC.".

*It is true that we have formulated this unclear and excessively long, especially at the beginning. We therefore changed from*

*'Our understanding of the sediment dynamics in the outer estuary and lower Ems River is primarily based on mathematical modelling and only limitedly based on in situ observations. Partly resulting from the absence of data, a number of key questions remain fuelled by lack of data or disagreement between earlier work. The first is that while transport mechanisms in the lower Ems River has been studied in great detail (see references above) less is known about the transport mechanisms in the channel connecting the lower Ems River and outer Ems estuary (the Emden Navigation Channel or ENC). The tides in the ENC are asymmetric with higher ebb flow velocities than flood flow velocities (Pein et al., 2014), and even though models result that spatial asymmetries are more important for residual transport than temporal asymmetries (Chernetsky et al., 2010) these tidal dynamics need to be explored in more detail. Also, salinity-driven flows appear insufficiently strong to import large quantities of suspended sediments landward (van Maren et al., 2015a). Secondly, some of the sediment in the lower Ems river is flushed seaward during periods of elevated discharge. The accumulation of sediment ion the lower Ems river may therefore lead to an increase in turbidity in the outer estuary. However, to what extent the high turbidity in the lower Ems river influences the turbidity in the outer Ems estuary (for instance during flushing) remains poorly known. And thirdly, the ENC requires large amounts of maintenance dredging, whereas from a hydrodynamic point of view it is one of the most energetic sections of the estuary. This points to the presence of convergence of suspended sediment transport, which in turn conflicts with the large dredging volumes in the lower Ems River suggesting up-estuary transport. '*

*Into*

*'But despite these recent advances in our knowledge on sediment dynamics within the lower Ems River and its estuary, three key questions remain related to the sediment dynamics. Firstly, we insufficiently understand how sediment is transported towards the lower Ems River.  The channel connecting the lower Ems River and outer Ems Estuary (the Emden Navigation Channel or ENC) are asymmetric with higher ebb flow velocities than flood flow velocities (Pein et al., 2014) and  salinity-driven flows appear insufficiently strong to import large quantities of suspended sediments landward (van Maren et al., 2015a). Secondly, to what extent the high turbidity in the lower Ems river influences the turbidity in the outer Ems Estuary (for instance during flushing) remains poorly known. And thirdly, the ENC requires large amounts of maintenance dredging, whereas from a hydrodynamic point of view it is one of the most energetic sections of the estuary. The only way to then explain the high siltation rates in such a dynamic area is strong convergence of suspended sediment transport. However, strong convergence of sediment transport in the ENC conflicts with the large up-estuary transport discussed above. '*

**Section 2**

This section discusses the design of the experiment. Subsections 2.1-2.3 do however mainly discuss processes and have some overlap with the introduction. It is not (always) clear why these sections are really needed for the design of the campaign (2.4). Subsection 2.4 is quite brief. Some more considerations on why these exact locations were chosen would help the reader. Why not a larger/smaller area of interest? Why this setting of moorings, frames and boats?

*We agree that the purpose of section 2.1-2.3 was unclear, even though they are very important. This is now made clear by adding an introduction at the beginning of section 2:*

*'The measurement campaign is designed to address knowledge gaps related to the sediment exchange between the lower Ems River and its estuary. These knowledge gaps have been summarized in three main research questions (introduced in the previous section). Designing a measurement campaign addressing these key questions requires an in-depth understanding of the relevant processes and associated time- and spatial scales. Therefore we will first elaborate the hydrodynamic and sedimentary processes associated with the three governing research questions (section 2.1-2.3), and subsequently translate this into an observation programme (section 2.4).'*

*Furthermore we have renamed the section titles to better reflect their purpose (Research Questions) and made a clearer connection from RQ's to monitoring programme (in 2.4) by replacing the sentence*

' It is likely that salinity-driven residual flows and mixing/stratification influence sediment exchange.'

*With a more thorough link between sections 2.1-2.3 and 2.4:*

'The previous evaluation of relevant processes reveals that water and sediment exchange is driven by the baroclinic processes resulting from salinity and SSC), barotropic tides, and low-frequency processes in particular the river discharge. The vertical exchange of sediment is influenced by mixing/stratification and flocculation processes. Sediment concentrations are very high, influencing sampling methodologies (turbidity but also flow velocity) and influencing processes (driving sediment-induced reduction of vertical mixing and horizontal flow velocities). The measurement campaign should therefore measure (1) the vertical structure of the water column over (2) periods covering a spring-neap tidal cycle and seasonal variations (especially related to the river discharge) and (3) a spatial domain covering parts of the lower Ems River (near Pogum) up to the outer Estuary and towards the Dollard, and (4) include processes related to mixing/stratification and flocculation. '

*And since we also agree that the sections 2.1-2.3 were excessively long (especially 2.1) we have removed the following section from 2.1:*

*'Regions of elevated sediment concentration within an estuary are referred to as Estuarine Turbidity Maxima (ETM's), and result from converging sediment transport mechanisms (see e.g. Burchard et al., 2017). These converging pathways are driven by river flow, estuarine circulation and lag effects (Dyer, 1994). Estuarine circulation is the combined effect of gravitational circulation (Postma, 1967), internal tidal asymmetry (Jay and Musiak, 1994) due to tidal straining (Simpson et al., 1990), lateral tidal residual flows (Lerczak and Geyer, 2004) and river flow; the relative importance of each component is strongly site-specific and often variable in time. Residual transport by time lag effects (such as settling lag and scour lag) are the result of finite values of the sediment properties (settling velocity, critical shear stress for erosion) in combination with asymmetries in the hydrodynamics (time or spatial asymmetries). Sediment properties vary throughout the tidal cycle, and – especially at high sediment concentrations – influence the hydrodynamics (diffusivity, viscosity).'*

**Section 3**

This section discusses the instrumentation. A clear overview is given. Also here, a few considerations would help. For example, why are the instruments on the mooring chains positioned at 1.5, 3.5 and ~7.8 m above the bed?

*We have (briefly) explained the motivation behind the instrumentation height (the 1.5 – 7.8 m MC's are for detecting vertical gradients; the RS frames focus on near-bed dynamics).*

Section 3.2 discusses the data processing. Outliers were removed, but how? Is the raw data stored as well (in case outliers might have been interesting data)? Can you indicate with a few lines how calibration of the OBSes was done and whether the calibration curves are in the data set?

*The reviewer correctly points out here and in section 4 that we provide too little information on data postprocessing prior to analysis. We have extended the sentence*

*'All data was carefully examined for outliers and spikes and, for the stationary surveys the OBS's were calibrated to physical units (SSC). '*

*to*

*'All data was carefully examined for outliers and spikes were removed manually and through filters (velocity data). ADV and ADCP data were filtered using the Signal-to-noise ratio of Elgar et al. (2005) and the 3D phase space method of (Goring and Nikora, 2002; Mori et al., 2007) – see also van Prooijen et al., 2020. The OBS's deployed in all permanent stations and onboard SB_EFW and SB_EMD were calibrated in the laboratory in two steps. First, the output of all OBS's were individually calibrated to NTU using a milk suspension. Secondly, one of the sensors was additionally calibrated against SSC, and this NTU – SSC relation is applied to the other sensors as well. '*

*With more information provided in section 4 (see hereafter)*

Line 211: "However, with the high concentrations in the Ems estuary acoustic and optical instruments become progressively less reliable, making accurate water sampling an important source of data. In order to minimise the potential impact of methodological errors all water samples were therefore analysed in the same laboratory.". This directly raises the question how (un)reliable the data are. In view of the use of the data by others, a reflection of the authors on this would help. In this paper, I would prefer that above an extensive interpretation of the data.

*Good point raised by the reviewer. We have added the following text in Chapter 4:*

*'The high SSC values only negatively influenced the ADCP measurements at the location Pogum (were concentrations were up to several kg/m3), which were therefore considered unreliable in both 2018 and 2019 (and therefore excluded from further processing). Other velocity measurements were not negatively impacted by high SSC because of the deployment of low-frequency ADCP's in areas with high SSC. The OBS calibration revealed that the SSC increase is slightly nonlinear with output voltage within the general range of SSC occurring in the study site (mostly up to several kg/m3) and therefore calibrated with a power function. The calibration remains linear to slightly non-linear up to 8 kg/m3; at higher SSC values the OBS output becomes unreliable. Such concentrations were only encountered at Pogum (or very infrequently at other locations). The point at which the output voltage starts decreasing with increasing SSC (as common for optical instruments) was not been reached during the field surveys. '*

**Section 4**

This section starts directly with interpretation of the data. Would it be possible to include a section on how well the instruments worked? For example, did OBSes reach their upper limit? Any biofouling? How (un)reliable were the acoustic and optic instruments in the high concentrations? Is a comparison made between samples and acoustic/optic instruments? What were the findings? See also line 211.

*The reviewer is completely right. We have renamed section 4 into 'results' rather than 'transport mechanisms 'added a new section 4.1:*

*4.1 Data collection*

*The majority of instrumentation worked well, with the following exception. The NTU values of the various OBS's were within 5% of each other except for 1 which is subsequently discarded from the dataset. The frame BM_KNO malfunctioned in 2019 for the complete period, while frame BM_GAT only collected data in the first 10 days of the 2019 deployment (see also Table 2). Frame BM_GAT also malfunctioned during the last 3 days of its 2018 deployment. LISST measurements onboard SB_KNO failed in both 2018 and 2019, and have therefore been excluded from this manuscript. No measurement errors resulting from sliding, slumping or boat accidents have been identified, and no biofouling was detected upon retrieval of the frames. The accuracy of the various instruments have not been investigated as part of this specific measurement campaign. However, decades of experiments with similar surveys, including dedicated accuracy tests, suggest that the accuracy of flow velocity observations is within 1%, discharges (using ADCP cross-section surveys) are within 5%, and SSC is within 10% (using OBS) to 20% (using ADCP).*

*The high SSC values only negatively influenced the ADCP measurements at the location Pogum (were concentrations were up to several kg/m3), which were therefore considered unreliable in both 2018 and 2019 (and therefore excluded from further processing). Other velocity measurements were not negatively impacted by high SSC because of the deployment of low-frequency ADCP's in areas with high SSC. The OBS calibration revealed that the SSC increase is slightly nonlinear with output voltage within the general range of SSC occurring in the study site (mostly up to several kg/m3) and therefore calibrated with a power function. The calibration remains linear to slightly non-linear up to 8 kg/m3; at higher SSC values the OBS output becomes unreliable. Such concentrations were only encountered at Pogum (or very infrequently at other locations). The point at which the output voltage starts decreasing with increasing SSC (as common for optical instruments) was not been reached during the field surveys.*

*We will highlight some of the key observations made during the EDoM campaigns illustrating the synoptic nature of the observations in a complex 3D flow environment with high suspended sediment concentrations, by examining residual flows and transport in more detail in the following sections (4.2 and 4.3) using velocity and ADCP observations. Additional turbulence data was collected at SB_KNO (August 2018 only) and at SB_EMD (both campaigns). The SB_KNO turbulence data provide insight in mixing and stratification processes in response to lateral and longitudinal flows (Schulz et al., 2020) whereas the SB_EMD data reveal how sediment-induced stratification processes may promote ebb-dominant sediment transport (Bailey et al., in prep.). The floc size measurements reveal a large variability in settling velocity within a tidal cycle (with higher settling velocities during the flood than during the ebb), but also a large seasonal variability: the settling velocity was higher during the August observations than during the January observations. Such a tidal variability may influence residual transport of sediment (promoting ebb transport) whereas the seasonal variation probably influences the seasonal variation in dredging (higher in the summer period).*

Section 4.3 is not balanced with the previous subsections, where the data is elaborated in more detail. Some of the findings have been published already apparently. It was also not directly clear where to find the floc data.

*Again correct. We have merged the old text*

*'Turbulence data was collected at SB_KNO (August 2018 only) and at SB_EMD (both campaigns). The SB_KNO turbulence data provide insight in mixing and stratification processes in response to lateral and longitudinal flows (Schulz et al., 2020) whereas the SB_EMD data reveal how sediment-induced stratification processes may promote ebb-dominant sediment transport (Bailey et al., in prep.). The floc size measurements reveal a large*

*variability in settling velocity within a tidal cycle (with higher settling velocities during the flood than during the ebb, but also a large seasonal variability: the settling velocity was higher during the August observations than during the January observations. '*

*With the new section 4.1.*

**Section 5**

Section 5.1 gives a clear overview of the findings. As indicated above, Section 5.2 mainly discusses the plans of the project team. Some of the mentioned plans for future research likely started already, as the data set is there for a couple of years now. This section could be modified to make the paper more attractive for other researchers. How could and why should researchers from outside the project team use the data set? Which type of data could be used for comparisons between estuaries for example? Which data can be used for intercomparison of instruments? What kind of things could not be analyzed by the research team, because a lack of experience?

*We have differentiated between recommendations that are fairly site-specific (the original recommendations, but now we mention explicitly that they are site-specific) and ones that are more generic, and made possible by this specific dataset. For this latter we have added the following text:*

*'These recommendations are site-specific, and some of these recommendations will be part of future research executed by the project partners. Nevertheless, we also invite researchers outside the project team to contribute to our understanding of sediment dynamics in the Ems Estuary. In addition to the site-specific data analysis directions provided above, the collected dataset also has the potential to advance our knowledge on*

- *Near-bed fine sediment dynamics measured with the RS_DOL and RS_EFW frames. Turbidity sensors were placed at 0.2, 0.3, 0.5 and 0.8 m above the bed providing valuable information, together with hydrodynamics (a downward-looking Aquadopp and near-bed ADV sensors) on near-bed fine sediment dynamics in turbid environments (which are limitedly understood – see van Maren et al., 2020).*
- *The use of optical and acoustic instruments for measuring SSC in high concentration environments, by comparing SSC values based on ADCP, OBS and water sample observations.*
- *Transverse flows and sediment transport patterns resulting from topographic constraints and density differences (joint analysis of the various mooring data, ship-borne moored observations and transect data collected in the ENC). '*

**Section 6**

The data availability section is not clear enough. It states that "most data" is stored on the repository. This directly raises the question why not all data is stored there, and which data is missing and why. Data has been averaged to 10-minute intervals, reducing the possibilities for analyses outside the project team. The statement in line 219: "We encourage use of the EDoM data by non-participants of the measurement campaign although all use of the data should be

communicated with the responsible surveyors." is a bit weird. I would recommend to open the data set completely, without such restrictions.

*We agree that the wording is a bit weird, as the reviewer points out. We also do not prohibit externals using the data from doing so. However, we strongly encourage all users to contact the persons that collected the data to jointly work with the data. Not only to provide the persons that put a lot effort in collecting the data with credits, but especially because with observational data details are often important. These details can only be provided by those who actually collected the data. And on the 10-minute averaging: this is a project agreement we made at the beginning of the project with all project partners. Jointly working with data collected by such a large and diverse group of people requires standardization, and we all agreed to deliver this type of data. We also believe that contacting the original collectors of the data is not a very big step anyhow. We have reformulated the original text reading*

*'Most data collected during the EDoM field campaign is stored on the repository of 4TU: https://doi.org/10.4121/c.6056564.v3 (van Maren et al., 2022). Most data stored on the repository is averaged at 10-minute intervals; water sample, settling velocity and turbulence data is stored at different intervals. For more details on the data itself, or access to the original (non-averaged) data, contact authors responsible for collection of the data of interest (see Table 2 for an overview of the responsible institute per measurement location). We encourage use of the EDoM data by non-participants of the measurement campaign although all use of the data should be communicated with the responsible surveyors. '*

*Into*

*'Most data collected during the EDoM field campaign is stored on the repository of 4TU: https://doi.org/10.4121/c.6056564.v3 (van Maren et al., 2022). Most data stored on the repository is averaged at 10-minute intervals; water sample, settling velocity and turbulence data is stored at different intervals. All data is freely available to all users. We do encourage anyone interested in using the data to contact the responsible surveyors (see Table 2 for an overview of the responsible institute per measurement location) for details on the data itself, but also to prevent multiple research groups to investigate similar topics in parallel. All data is averaged to 10-minute average values for standardisation purpose and easy access. The original (non-averaged) data may be acquired by contacting authors responsible for collection of the data of interest.'*

The data on the repository is divided over different directories. It is now a bit difficult to find the observation stations from Table 2 in the repository. Adding the directories in Table 2 might help?

*We prefer to keep the organization of the 4TU website separate from the paper, to allow restructuring of the repository in a later stage (when at that time deemed relevant). Again, we also encourage potential data users to contact he authors (even though we did our best to make everything as clear as possible for all potential users).*